# A Spectral Theory of Neural Prediction and Alignment

**Abdulkadir Canatar**[1,2*]    **Jenelle Feather**[1,2*]    **Albert J. Wakhloo**[1,3]    **SueYeon Chung**[1,2]

[1]Center for Computational Neuroscience, Flatiron Institute
[2]Center for Neural Science, New York University
[3]Zuckerman Institute, Columbia Univeristy
[*]Equal contribution
`{acanatar,jfeather,awakhloo,schung}@flatironinstitute.org`

## Abstract

The representations of neural networks are often compared to those of biological systems by performing regression between the neural network responses and those measured from biological systems. Many different state-of-the-art deep neural networks yield similar neural predictions, but it remains unclear how to differentiate among models that perform equally well at predicting neural responses. To gain insight into this, we use a recent theoretical framework that relates the generalization error from regression to the spectral properties of the model and the target. We apply this theory to the case of regression between model activations and neural responses and decompose the neural prediction error in terms of the model eigenspectra, alignment of model eigenvectors and neural responses, and the training set size. Using this decomposition, we introduce geometrical measures to interpret the neural prediction error. We test a large number of deep neural networks that predict visual cortical activity and show that there are multiple types of geometries that result in low neural prediction error as measured via regression. The work demonstrates that carefully decomposing representational metrics can provide interpretability of how models are capturing neural activity and points the way towards improved models of neural activity.

## 1   Introduction

Neuroscience has a rich history of using encoding models to understand sensory systems [1, 2, 3]. These models describe a mapping for arbitrary stimuli onto the neural responses measured from the brain and yield predictions for unseen stimuli. Initial encoding models were built from hand-designed parameters to model receptive fields in early sensory areas [4, 5], but it was difficult to extend these hand-engineered models to capture responses in higher-order sensory regions. As deep neural networks (DNNs) showed human-level accuracy on complex tasks, they also emerged as the best models for predicting brain responses, particularly in these later regions [6, 7, 8, 9, 10].

Although early work showed that better task performance resulted in better brain predictions [6], in modern DNNs, many different architectures and training procedures lead to similar predictions of neural responses [11, 12, 13]. This is often true even when model representations are notably different from each other using other metrics of comparison [14, 15]. For instance, previous lines of work have shown clear differences between models from their behavioral responses to corrupted or texturized images [16, 17, 18], from their alignment to human similarity judgments [19], from generating stimuli that are "controversial" between two models [20], and from stimuli that are metameric to models [15] or to human observers [21]. Other work has focused on directly comparing representations [22, 23] or behavioral patterns [24] of one neural network to another, finding that changing the training dataset or objective results in changes to the underlying representations. Given the increasing number of findings that demonstrate large variability among candidate computational models, it is an open

question of why current neural prediction benchmarks are less sensitive to model modification, and how to design future experiments and stimulus sets to better test our models.

Metrics of similarity used to evaluate encoding models of neural responses are often condensed into a single scalar number, such as the variance explained when predicting held-out images [6, 11]. When many models appear similar with such metrics, uncovering the underlying mechanisms driving high predictivity is a challenge [25]. To uncover the key factors behind successful DNN models of neural systems, researchers have proposed various approaches to explore the structural properties of internal representations in DNN models in relation to neural data. One promising approach focuses on analyzing the geometry of neural activities [25]. Several studies have examined the evaluation of geometries and dimensionalities of DNNs [26, 27, 28], highlighting the value of these geometric approaches. They provide mechanistic insights at the population level, which serves as an intermediate level between computational level (e.g., task encoding) and implementation level (e.g., the activities of single neurons or units). Furthermore, the measures derived from representation geometries can be utilized to compare the similarity between models and neural data [7, 25, 29]. Although these approaches have been successful, they do not directly relate the geometry of DNNs to the regression model used to obtain neural predictions, which uniquely captures the utility of a model for applications such as medical interventions [11] and experimental designs [30]. Recent empirical findings by Elmoznino et al. [31] have suggested a correlation between the high-dimensionality of model representations and neural predictivity in certain scenarios, but this work lacks explicit theoretical connections between the spectral properties and predictivity measures. Our work contributes to this line of investigation, by seeking a theoretical explanation and suggesting new geometric measures linked to a model's neural prediction error for future investigations.

To bridge the gap between DNN geometry and encoding models used for neural prediction, we turn to a recent line of work that derived a predictive theory of generalization error using methods from statistical physics [32, 33]. Specifically, two different spectral properties that affect the regression performance were identified: *Spectral bias* characterizing how fast the eigenvalues of the data Gram matrix decay, and *task-model alignment* characterizing how well the target aligns with the eigenvectors of this data Gram matrix [33]. While the former describes how large the learnable subspace is, the latter defines how much of the variance of the target is actually learnable. Note that in general, these two properties may vary independently across representations. Hence, the trends among regression scores across DNNs must be carefully studied by taking into account the joint trends of spectral bias and task-model alignment.

In this work, we explore how encoding models of neural responses are influenced by the geometrical properties of the DNN representations and neural activities. Our analyses primarily concern how and why models achieve the predictivities that they do, rather than offering an alternative scalar metric for predictivity. We find that the neural prediction error from ridge regression can be broken down into the radius and dimension of the *error mode geometry*, characterizing the error along each eigenvector. When investigating models along these error axes, we see a large spread in the error mode geometry, suggesting that the correlation values obtained from regression analyses obscure many geometrical properties of the representations.

Our primary contributions are:

- We analytically decompose the neural prediction error of ridge regression from a model to a set of brain data in terms of the model eigenspectra, the alignment between the brain data and the eigenvectors of the model, and the training set size.

- We introduce two geometric measures that summarize these spectral properties and directly relate to the neural prediction error. We show that these measures distinguish between different models with similar neural prediction errors using a wide variety of network architectures, learning rules, and firing rate datasets from visual cortex.

- Using spectral theory, we demonstrate that for networks effective in predicting neural data, we can ascertain if their superior performance stems from the model's spectra or alignment with the neural data. Our findings indicate that (a) trained neural networks predict neural data better than untrained ones due to better alignment with brain response and (b) adversarial training leads to an improved alignment between the eigenvectors of networks and early visual cortex activity.

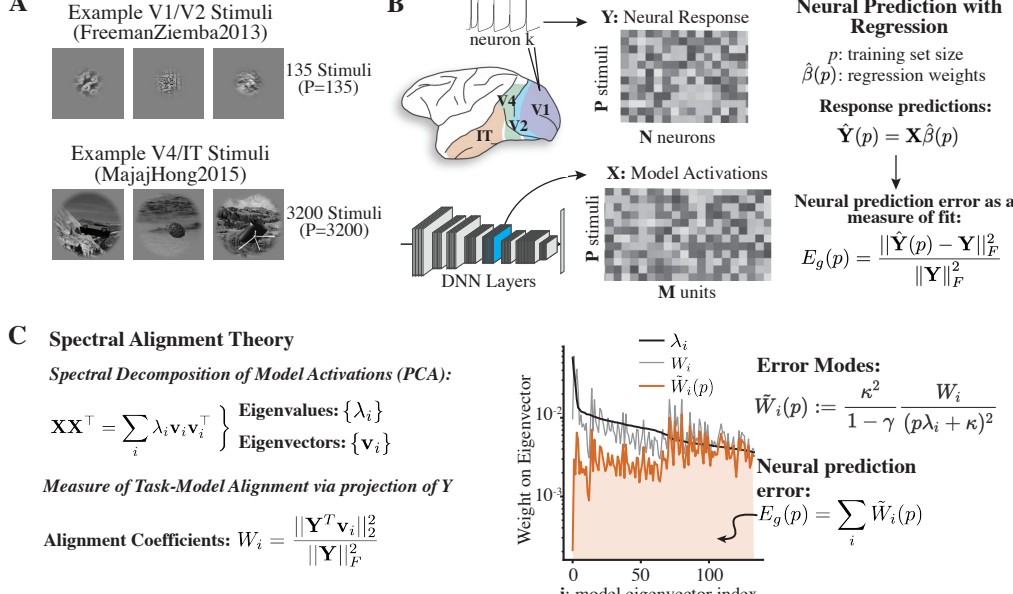

Figure 1: **Experimental and theoretical framework. (A)** Example stimuli from visual experiments used for neural prediction error measures of V1, V2, V4, and IT. **(B)** Framework for predicting neural responses from model activations. Responses are measured from a brain region $\mathbf{Y}$ and from a model stage of a DNN $\mathbf{X}$. Weights for linear regression are learned to map $\mathbf{X}$ to $\mathbf{Y}$, and the empirical neural prediction error $E_g(p)$ is measured from the response predictions. **(C)** Spectral alignment theory. Neural prediction error can be described as a function of the eigenvalues and eigenvectors of the model Gram matrix. The neural response matrix $\mathbf{Y}$ is projected onto the eigenvectors $\mathbf{v}_i$ with alignment coefficients $W_i$. For the number of training samples $p$, the value of each error mode is given as $W_i E_i(p) = \widetilde{W}_i(p)$. The total neural prediction error can be expressed as a sum of $\widetilde{W}_i(p)$ terms, visualized as the area under the $\widetilde{W}_i(p)$ curve.

## 2 Problem Setup

### 2.1 Encoding Models of Neural Responses

We investigated DNNs as encoding models of visual areas V1, V2, V4, and IT. The neural datasets were previously collected on 135 texture stimuli for V1 and V2 [34], and on 3200 object stimuli with natural backgrounds for V4 and IT [35] (Fig. 1A). Both datasets were publicly available and obtained from [11]. The neural responses and model activations were extracted for each stimulus (Fig. 1B), and each stage of each investigated neural network was treated as a separate encoding model. We examined 32 visual deep neural networks. Model architectures included convolutional neural networks, vision transformers (ViTs), and "biologically inspired" architectures with recurrent connections, and model types spanned a variety of supervised and self-supervised training objectives (see Sec. SI.2 for full list of models). We extracted model activations from several stages of each model. For ViTs, we extracted activations from all intermediate encoder model stages, while in all other models we extracted activations from the ReLU non-linearity after each intermediate convolutional activation. This resulted in a total number of 516 analyzed model stages. In each case, we flattened the model activations for the model stage with no subsampling.

### 2.2 Spectral Theory of Prediction and Alignment

In response to a total of $P$ stimuli, we denote model activations with $M$ features (e.g. responses from one stage of a DNN) by $\mathbf{X} \in \mathbb{R}^{P \times M}$ and neural responses with $N$ neurons (e.g. firing rates) by $\mathbf{Y} \in \mathbb{R}^{P \times N}$. Sampling a training set $(\mathbf{X_{1:p}}, \mathbf{Y_{1:p}})$ of size $p < P$, ridge regression solves (Fig. 1B):

$$\hat{\beta}(p) = \underset{\beta \in \mathbf{R}^{M \times N}}{\arg \min} ||\mathbf{X_{1:p}}\beta - \mathbf{Y_{1:p}}||_F^2 + \alpha_{\text{reg}}||\beta||_F^2, \quad \hat{\mathbf{Y}}(p) = \mathbf{X}\hat{\beta}(p) \tag{1}$$

We analyze the normalized neural prediction error, $E_g(p) = \frac{||\hat{\mathbf{Y}}(p) - \mathbf{Y}||_F^2}{||\mathbf{Y}||_F^2}$, for which we utilize theoretical tools from learning theory [36, 37, 38], random matrix theory [39, 40, 41] and statistical physics [42, 32, 33] to extract geometrical properties of representations based on spectral properties of data. In particular, the theory introduced in [32, 33] relies on the orthogonal mode decomposition (PCA) of the Gram matrix $\mathbf{X}\mathbf{X}^\top$ of the model activations, and projection of the target neural responses onto its eigenvectors:

$$\mathbf{X}\mathbf{X}^\top = \sum_{i=1}^{P} \lambda_i \mathbf{v}_i \mathbf{v}_i^\top, \quad W_i := \frac{||\mathbf{Y}^T \mathbf{v}_i||_2^2}{||\mathbf{Y}||_F^2}, \quad \langle \mathbf{v}_i, \mathbf{v}_j \rangle = \delta_{ij}. \tag{2}$$

Here, associated to each mode $i$, $W_i$ denotes the variance of neural responses $\mathbf{Y}$ in the direction $\mathbf{v}_i$, and $\lambda_i$ denotes the $i^{th}$ eigenvalue. Then, the neural prediction error is given by:

$$E_g(p) = \sum_{i=1}^{P} W_i E_i(p), \quad E_i(p) := \frac{\kappa^2}{1-\gamma} \frac{1}{(p\lambda_i + \kappa)^2}, \tag{3}$$

where $\kappa = \alpha_{\text{reg}} + \kappa \sum_{i=1}^{P} \frac{\lambda_i}{p\lambda_i + \kappa}$ must be solved self-consistently, and $\gamma = \sum_{i=1}^{P} \frac{p\lambda_i^2}{(p\lambda_i + \kappa)^2}$ (see Sec. SI.1 for details) [32, 33]. Note that the theory depends not only on the model eigenvalues $\lambda_i$, but also on the model eigenvectors $\mathbf{v}_i$ along with the responses $\mathbf{Y}$, which determine how the variance in neural responses distributed among a model's eigenmodes.

Although the equations are complex, the interpretations of $W_i$ and $E_i(p)$ are simple. $W_i$ quantifies the projected variance in neural responses on model eigenvectors (alignment between neural data and model eigenvectors, i.e., *task-model alignment* [33]). Meanwhile, $E_i(p)$, as a function of the training set size $p$, determines the reduction in the neural prediction error at mode $i$ and depends only on the eigenvalues, $\lambda_i$ (i.e., *spectral bias* [33]).

In this work, we combine both and introduce *error modes* $\widetilde{W}_i(p) := W_i E_i(p)$:

$$\widetilde{W}_i(p) := \frac{\kappa^2}{1-\gamma} \frac{W_i}{(p\lambda_i + \kappa)^2}, \quad E_g(p) = \sum_i \widetilde{W}_i(p) \tag{4}$$

As shown in (Fig. 1C), $\widetilde{W}_i$ associated with large eigenvalues $\lambda_i$ will decay faster with increasing $p$ than those associated with small eigenvalues.

The generalization performance of a model is fully characterized by its error modes, $\widetilde{W}_i$. However, due to its vector nature, $\widetilde{W}_i$ is not ideally suited for model comparisons. To address this limitation, we condense the overall shape of $\widetilde{W}_i$ into two geometric measures, while preserving their direct relationship to the neural prediction error. This is in contrast to previous such measures, such as the effective dimensionality, that only depend on model eigenvalues [31].

Here, we define a set of geometric measures that characterize the distribution of a model's $\widetilde{W}_i$ via the *error mode geometry* (Fig. 2A). Specifically, we rewrite the neural prediction error $E_g(p)$ as:

$$R_{em}(p) := \sqrt{\sum_i \widetilde{W}_i(p)^2}, \quad D_{em}(p) := \frac{\left(\sum_i \widetilde{W}_i(p)\right)^2}{\sum_i \widetilde{W}_i(p)^2}, \quad E_g(p) = R_{em}(p)\sqrt{D_{em}(p)}. \tag{5}$$

The error mode radius $R_{em}$ denotes the overall size of the error terms, while the error mode dimension $D_{em}$ represents how dispersed the total neural prediction error is across the different eigenvectors (Fig. 2A). Note that, the neural prediction error $E_g(p)$ above has a degeneracy of error geometries; many different combinations of $R_{em}$ and $D_{em}$ may result in the same $E_g(p)$ (Fig. 2B). Given a fixed neural prediction error, a higher value of $R_{em}(p)$ indicates that only a few $\widetilde{W}_i$ are driving the error, while a higher $D_{em}(p)$ indicates that many different $\widetilde{W}_i$ are contributing to $E_g(p)$.

## 3 Results and Observations

We first confirmed that Eq. 3 accurately predicted the neural prediction error for the encoding models considered here. We performed ridge regression from the model activations of each model stage of

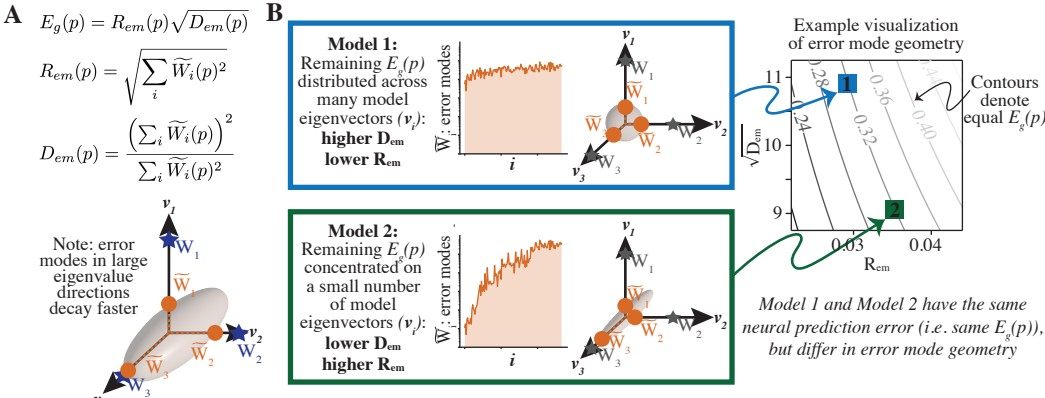

Figure 2: **Summary measures for error mode geometry.** **(A)** Error modes ($\widetilde{W}_i$) are related to the model eigenspectra ($\lambda_i$) and alignment coefficients ($W_i$). Error mode geometry summarized by $R_{em}$ and $D_{em}$ characterizes the distribution of error modes and directly relates to the neural prediction error $E_g(p)$ **(B)** Interpreting contour plots. Error mode geometry distinguishes between models with the same prediction error ($E_g(p)$). Neural prediction error is the product of $R_{em}$ and $\sqrt{D_{em}}$ and is thus easily visualized on a contour plot where each contour represents equal $E_g(p)$.

each trained neural network to the neural recordings for each cortical region using a ridge parameter of $\alpha_{\text{reg}} = 10^{-14}$, and also calculated the theoretical neural prediction error for each using Eq. 3. Given that Eq. 3 is only accurate in the large $P$ limit in which both the number of training points and the number of test samples is large (see Sec. SI.1 and Fig. SI.4.2), we used a 60/40 train-test split ($p = 0.6P$) in the experiments below. As shown in Fig. 3A, this split yielded near-perfect agreement between the theoretical and empirical neural prediction errors for areas V4 and IT ($R^2$=0.97 and 0.97, respectively), and very strong agreement for areas V1 and V2 ($R^2$=0.9 and 0.9, respectively). We present further data for this agreement in Sec. SI.4. Furthermore, we found that the ordering of models according to their neural prediction error is very similar to the ordering obtained via the partial least squares (PLS) regression method used in Brain-Score [11] (see Sec. SI.3). We maintained these choices for train-test split sizes and $\alpha_{\text{reg}}$ for all subsequent experiments.

We visualized the spread of error mode geometries for each brain region across different trained models in Fig. 3B. Each point is colored by the empirical $E_g(p)$, while the contour lines show the theoretical $E_g(p)$ value. Among trained models, there was a range of $R_{em}$ and $D_{em}$ values even when many models have similar $E_g(p)$. This demonstrates that our geometrical interpretation of the error mode geometry can give additional insights into how models achieve low neural prediction error. In the following sections, we explore how these geometrical properties of the neural prediction error vary with model stage depth, training scheme, and adversarial robustness.

### 3.1 Error Mode Geometry Varies across Model Stages

We analyzed the neural prediction error obtained from activations at different model stages. In Fig. 4A, we plot the error mode geometry for all analyzed DNN model stages when predicting brain data from V1 and IT. Each point is color coded based on its relative model depth. Lower neural prediction errors were achieved (i.e. points lie on lower $E_g$ contours) for earlier model stages in early visual area V1, and for later model stages in downstream visual area IT. This observation is in agreement with previous findings [6, 11, 43] where artificial neural networks have a similar hierarchy compared to the visual cortex. Qualitatively, the trends across model stages were more prominent than differences observed from comparing model architectures for supervised models (Fig. SI.5.1), or from comparing supervised trained models to self-supervised models of the same architecture (Fig. SI.5.2).

To investigate the source of this geometric difference across model stages, we performed an experiment on each model where we took the eigenspectra $\lambda_i$ measured from the first (Early) or last (Late) model stage and paired this with the alignment coefficients $W_i$ measured from the Early or Late model stage (Fig. 4B). These two spectral properties can be varied independently, as $W_i$ depends on

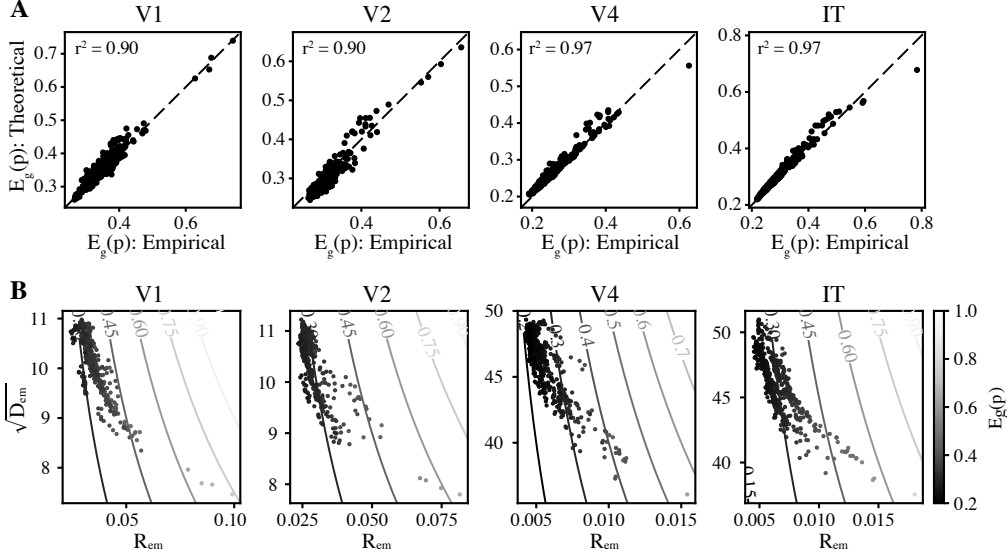

Figure 3: **Error mode geometry of model predictions for each brain region.** (A) Theoretical values of neural prediction error match empirical values from ridge regression. Each point corresponds to predictions obtained from one of the 516 analyzed model stages. (B) Error mode dimension and radius for each brain region. Each point corresponds to the error mode $\sqrt{D_{em}}$ and $R_{em}$ for the region from a specific model stage of trained neural network models. Points are colored by the empirical error from performing ridge regression between the model activations and neural responses. $R_{em}$ and $\sqrt{D_{em}}$ values are obtained from theoretical values, and contour lines correspond to theoretical values of the neural prediction error, such that points along the contours have the same error.

the model eigenvectors but not on the model eigenvalues. We then measured the neural prediction error that would be obtained from such a model. This experiment allows us to isolate whether the observed differences in neural prediction error are primarily due to the $\lambda_i$ or $W_i$ terms. In the V1 data, there was little difference in the error mode geometry between the location of the $W_i$ terms, however using early model stage $\lambda_i$ corresponds to lower neural prediction error (summarized in Fig. 4C). In contrast, for the IT data using the late stage $W_i$ terms corresponded lower $E_g(p)$ mainly through a reduction of the error mode $R_{em}$ (i.e. many error modes are better predicted). The spectral properties are shown in full for the early and late stages of ResNet50 as measured on the V1 and IT datasets (Fig. 4D). These results highlight the need to study the spectral properties of the model together with the brain responses as summarized by $\widetilde{W_i}$: Simply studying the properties of the eigenspectra of the model is insufficient to characterize neural prediction error (see SI.5.5 for more details).

## 3.2 Trained vs. Untrained

We analyzed how the error mode geometry for neural predictions differed between trained and randomly initialized DNNs (Fig. 5A). In line with previous results [11, 14, 44], we found that training yielded improved neural predictions as measured via smaller $E_g(p)$ (Fig. 5B). This improvement was most notable in regions V2, V4, and IT, where there was also a characteristic change in the error mode geometry. In these regions, while $R_{em}$ decreased with training, $D_{em}$ surprisingly increased. However, the decrease in $R_{em}$ was large enough to compensate for the higher $D_{em}$, leading to an overall reduction in $E_g(p)$. In V1, we observed only a small difference in neural prediction error between trained and untrained models, similar to previous results for early sensory regions [45, 14]. As shown in Fig. SI.5.7, we found qualitatively similar results in a different V1 dataset with a larger number of tested stimuli [46].

What differed between the trained and random model representations that led to differences in error mode geometry? To gain insight into this, we investigated how the eigenspectra ($\lambda_i$) and the alignment coefficients ($W_i$) individually contributed to the observed error mode geometry in visual area IT. We performed an experiment on the trained and random ResNet50 model activations where

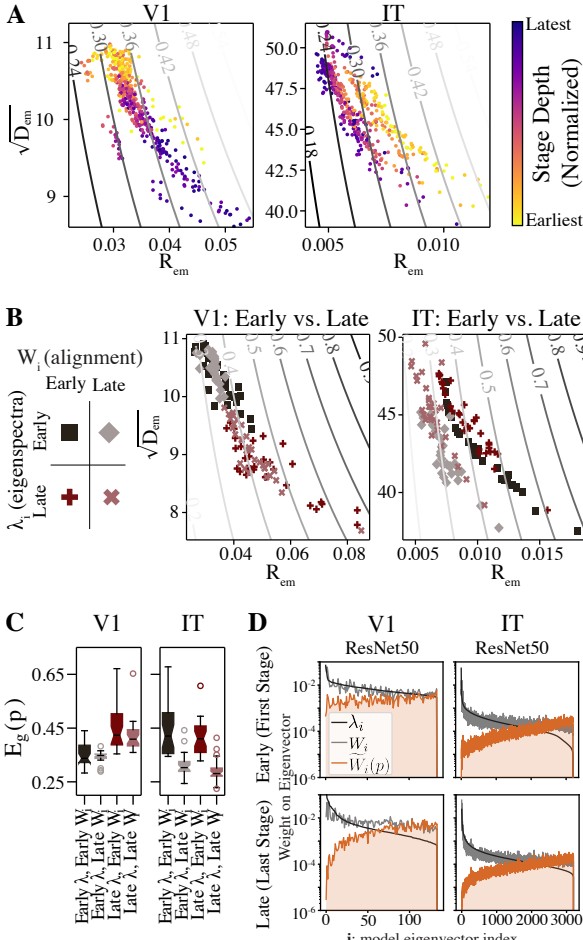

Figure 4: **Model stage depth and error mode geometry.** **(A)** Each point represents the error mode geometry obtained from a specific model stage of a trained neural network, where the color of the value represents the depth of the model stage. The color is normalized for each model to the $[0, 1]$ range where the earlier and later model stages have lighter and darker colors, respectively. $R_{em}$ and $\sqrt{D_{em}}$ values are obtained from theoretical values, and contour lines correspond to theoretical values of the neural prediction error as in Fig. 2. **(B,C)** Predicted $E_g(p)$ that would be obtained when using the eigenspectra ($\lambda_i$) from the first (Early) or last (Late) stage of each of the 32 analyzed models, paired with the alignment coefficients ($W_i$) terms from the Early or Late stage of each model. Full error mode geometry is given in (B) where each point corresponds to a single model with the chosen $W_i$ and $\lambda_i$ terms, and (C) shows box plots summarizing the $E_g(p)$ from each comparison. In V1, using the $\lambda_i$ from the early stage of the model resulted in better neural prediction error and a lower $R_{em}$ but higher $\sqrt{D_{em}}$, regardless of the $W_i$ terms used. In IT, using the alignment terms $W_i$ from the late model stage resulted in a smaller neural prediction error and lower $R_{em}$, with little change in $\sqrt{D_{em}}$. **(D)** Full spectra for $\lambda_i$, $W_i$ and $\widetilde{W}_i$ from Early and Late model stages of ResNet50 on the V1 and IT datasets.

we measured $\lambda_i$ from one model and paired it with the $W_i$ measured from the other model (Fig. 5C). Using $W_i$ from the trained model led to much lower $R_{em}$ than when using $W_i$ from the random model. This decrease in $R_{em}$ when using the $W_i$ terms from the trained model was the main driver of the improvement in $E_g(p)$ when we used the trained model to predict the IT neural data. In contrast, when using the eigenspectra of the random model, the $D_{em}$ was lower than when using the eigenspectra of the trained model, at the cost of a slight increase in $R_{em}$. Note that the model eigenspectra clearly decayed at a significantly faster rate in the random vs. trained models (Fig. 5D), but it was the properties of the alignment terms $W_i$ that drove the change in $E_g(p)$ between trained and untrained models. Thus, the eigenvectors of the trained model were better aligned with the neural data compared to the random model, which resulted in low prediction error regardless of which eigenspectra was used.

### 3.3    Adversarial vs. Standard Training

Recent work has suggested that adversarially trained networks have representations that are more like those of biological systems [47, 48, 15]. We analyzed the neural prediction error and error mode geometry of adversarially trained ("Robust") neural network models [49, 50], and their non-adversarially trained counterparts. The error mode geometry for an $\ell_2(\epsilon = 3)$ ResNet50 is given in Fig. 6A, and similar results are shown for an $\ell_\infty(\epsilon = 4)$ ResNet50 in Fig. SI.5.8.

The effect of adversarial training on error mode geometry varied across model stages and cortical regions. In the V1 dataset, lower neural prediction error was observed for early stages of robust

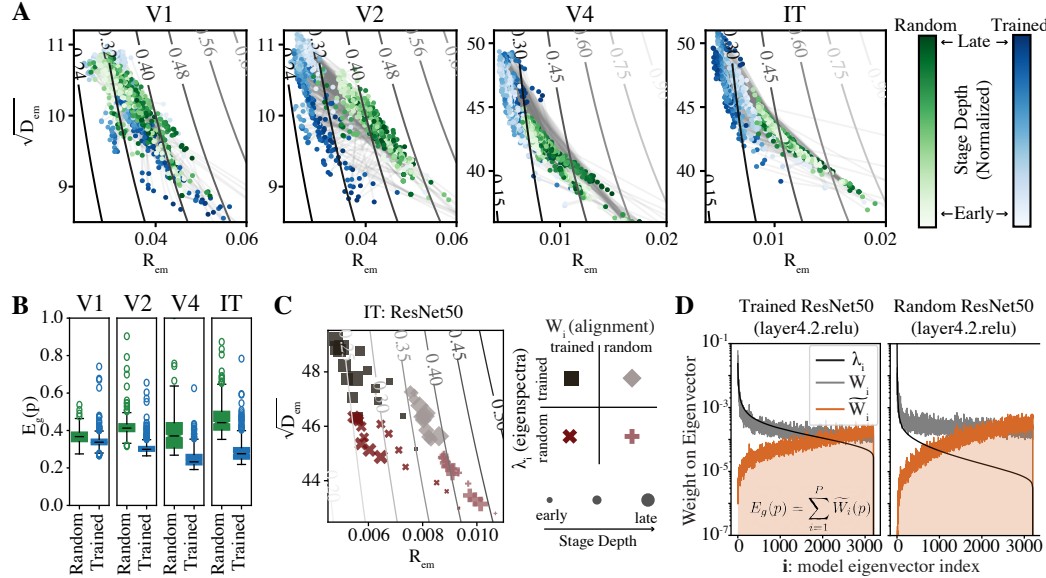

Figure 5: **Error mode geometry changes with model training. (A)** Error mode for trained models (blue) and randomly initialized models (green) for each brain region. The same model stage of each network is connected with a line, and stage depth is shown with opacity. Contours are given by theoretical values of $E_g(p)$, as in previous plots. **(B)** Neural prediction error was lower for trained models compared to random models in all brain regions, but most notable in V2, V4, and IT. Median is given as a black line and box extends from the first quartile to third quartile of the data. Whiskers denote 1.5x the interquartile range. **(C)** Predicted $E_g(p)$ that would be obtained on IT dataset when using eigenspectra from trained or random ResNet50 models, paired with the alignment $W_i$ terms from each of the trained or random ResNet50 models. Size of point denotes model stage depth. Using the $W_i$ terms from the trained model resulted in a lower $E_g(p)$ due to a lower $R_{em}$, while using the $\lambda_i$ from the trained models increased $D_{em}$ and did not appreciably change $E_g(p)$. **(D)** Model eigenvalues, alignments, and error modes for the final analyzed model stage of the ResNet50 architecture on IT dataset for both trained and random models. Even though the eigenspectra for the random model was notably different than the eigenspectra for the trained model, we observed in (C) that the driver of the decreased $E_g(p)$ for the trained model was a difference in the $W_i$ terms.

models (Fig. 6B), in line with previous work [48]. This corresponded to a decrease in $R_{em}$ with little change in $D_{em}$, suggesting that the decrease in neural prediction error was shared relatively equally across all error modes. In contrast, at late model stages, $D_{em}$ was smaller for the robust models compared to the standard models, but there was no difference in neural prediction error between the robust and standard networks. In regions V2, V4, and IT, we observed that there was little difference in neural prediction error between the standard and robust models, but that error mode geometry in robust models was associated with a lower $D_{em}$ and a higher $R_{em}$. This suggests that the difference between robust and standard models in these regions was due to a small number of error modes having higher $\widetilde{W_i}$ in the robust models. This did not lead to better neural prediction error for regions V2, V4, and IT, but nevertheless, the error mode geometry revealed that the data was predicted in different ways.

To better understand the differences between robust and standard networks in V1, we ran a series of experiments testing the effects of adversarial training on model eigenspectra and alignment with neural data. For each model stage of the ResNet50 architecture, we paired the eigenspectra ($\lambda_i$) of the standard or robust network with the alignment coefficients ($W_i$) of the standard or robust network (Fig. 6C). We then calculated the error mode geometry and neural prediction error for all four possible pairings. This approach allowed us to examine how adversarial training-related changes in the spectrum and alignment coefficients affected the error mode geometry for each model stage.

We found that the alignment coefficients of the robust models consistently yielded lower V1 prediction error (Fig. 6D). In early model stages, using the $W_i$ from the robust models led to a lower neural

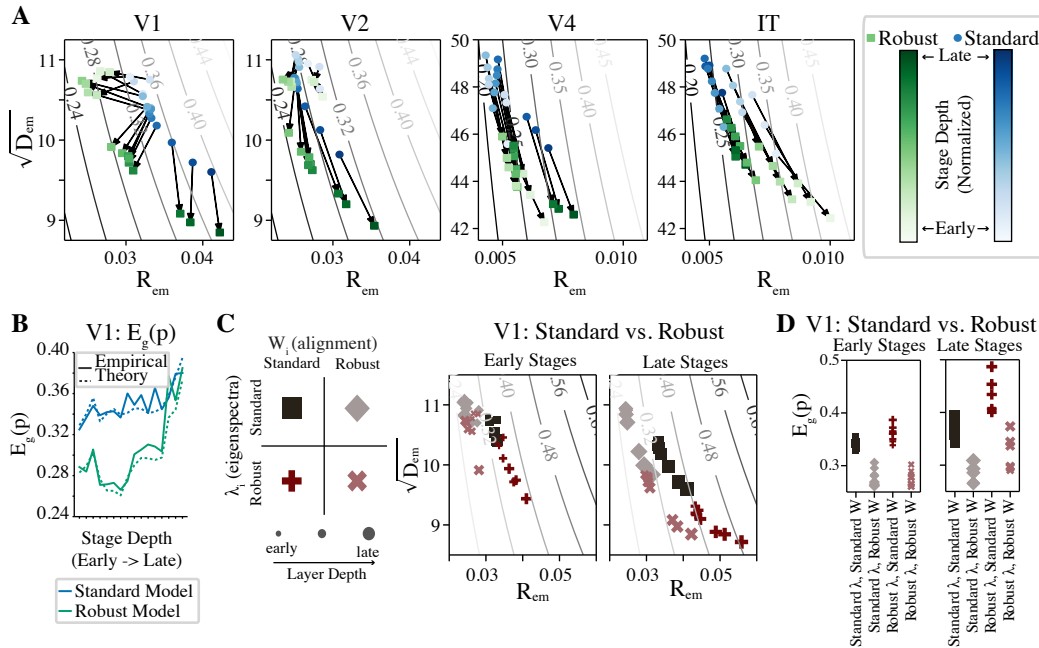

Figure 6: **Error mode geometry and adversarial robustness.** **(A)** Error mode geometry for standard ResNet50 (Blue) and adversarially-trained (Robust) ResNet50 (Green) models for each brain region. The same stage of each network is connected with an arrow, and stage depth is shown with opacity. Contours are given by theoretical values of $E_g(p)$ as in previous plots. **(B)** For standard and robust ResNet50 models, $E_g(p)$ for V1 data as a function of stage depth. **(C)** Predicted $E_g(p)$ that would be obtained on V1 dataset for $\lambda_i$ of standard or robust ResNet50 networks paired with $W_i$ of standard or robust ResNet50 networks. Each point was from a single stage of the model, and plots are broken up by the early stages (first 8/16 analyzed stages) and the late stages (last 8/16 analyzed stages) of the model for visualization purposes. **(D)** Summarized neural prediction error for the experiment in (C). For all stages, the robust $W_i$ led to lower error, but in the late stages, there was an additional effect from the choice of $\lambda_i$.

prediction error via reduced $R_{em}$, and the choice of $\lambda_i$ had a negligible effect on the prediction error for these stages. In the later model stages, the $W_i$ from the robust models also achieved a lower $E_g$, but for these stages, the choice of $\lambda_i$ had a non-negligible impact on the resulting prediction error. Overall, these results suggest that adversarial training leads to an improved alignment between the eigenvectors of the model and early visual cortex activity.

## 4   Discussion

Many metrics have been proposed to compare model representations to those observed in biological systems [51, 52, 22, 23, 11]. However, it is typically unclear which aspects of the representational geometry lead to a particular similarity score. In this work, we applied the theory of spectral bias and task-model alignment for kernel regression [33] to the generalization error for encoding models of neural activity. This provides a theoretical link between a model's ability to predict neural responses and the eigenspectra and eigenvectors of model representations. We introduced geometric measures $R_{em}$ and $D_{em}$ of the error modes to summarize how the spectral properties relate to neural prediction error, and showed that models with similar errors may have quite different geometrical properties. We applied this theory to investigate the roles of stage depth, learned representations, and adversarial training in neural prediction error throughout the ventral visual stream.

### *Dimensionality of neural and model representations*

Recent work has investigated spectral properties of neural activity in biological [53, 54] and artificial [31, 28] systems. One empirical study suggested that high-dimensional model representations have

lower neural prediction error, where effective dimensionality was defined as the participation ratio of the eigenvalues alone [31]. However, we empirically found that this effective dimensionality of representations was not well correlated with prediction error for our datasets (Sec. SI.5.5). The lack of a consistent relationship between effective dimensionality and neural prediction error was expected based on our theory, as the neural prediction error depends both on the eigenvalues and eigenvectors of a model. These quantities co-vary from model to model, and thus model comparisons should include information from both. In this work, we demonstrated how our theory can be used to examine whether differences in observed prediction error are primarily due to the eigenvalues or alignment coefficients (which are dependent only on the eigenvectors), and found that in many instances, the alignment coefficients were the main driver of lower prediction error.

### Dependence of Neural Prediction Error on Training Set Size

Our spectral theory of neural prediction characterizes the relationship between the training set size, the generalization error, and the spectral properties of the model and brain responses. This highlights that model ranking based on neural prediction error may be sensitive to the train/test split ratio. In Eq. 3, we see that, for small $p$, the error modes corresponding to small $\lambda_i$ generally remain close to their initial value ($\widetilde{W}_i(p) \approx \widetilde{W}_i(0)$), and hence their contribution to neural prediction error does not decay (Eq. 4). On the other hand, at large sample size $p$, these small $\lambda_i$ error modes change from their initial values and therefore may change the ranking of the models. These observations regarding the role of model eigenspectra and dataset size may be used in the future to design more comprehensive benchmarks for evaluating models.

### Limitations

Although our empirical analysis follows what we expect from theory and shows a diversity of error mode geometries, the datasets that we used (particularly for V1 and V2) have a limited number of samples. In some cases, such as the large $p$ limit, theoretical results for the neural prediction error may not match empirical results. Additionally, we only investigated one dataset for V2, V4, and IT, and only two datasets for V1. Future work may reveal different trends when using different datasets. Moreover, this theory assumes that the brain responses are deterministic functions of the stimuli. As shown in [33], this assumption may be relaxed, and future work can examine the role of neuronal noise in these contexts.

### Future work

The results in this paper demonstrate that our theory can characterize the complex interplay between dataset size, representational geometry, and neural prediction error. It provides the basis for numerous future directions relating spectral properties of the model and data to measures of brain-model similarity. Insights from our theory may suggest ways that we can build improved computational models of neural activity.

While we focused on DNN models pre-trained on image datasets, a closely related line of work has investigated the properties of DNN models directly trained to predict neural responses [46, 55, 56]. Future work could compare the error mode geometries from end-to-end networks to those obtained from networks pre-trained on image data. Such an analysis could elucidate how these two vastly different training schemes are both able to predict neural data well. Additionally, our experiments focused on the case of the same small regularization coefficient for all regression mappings, but future work could explore the case of optimal regularization. The spectral theory can also be applied to other types of brain-model comparisons, such as Partial Least Squares or Canonical Correlation Analysis.

Although we were able to differentiate many error mode geometries based on $R_{em}$ and $\sqrt{D_{em}}$, we initially expected to see many more values of $(R_{em}, \sqrt{D_{em}})$ for each fixed $E_g(p)$. The investigated vision models seem to be constrained to a particular region in the $(R_{em}, \sqrt{D_{em}})$-space (Fig. 3). Future work may shed light on whether this is due to implicit biases in model architectures or due to a fundamental coupling of ($R_{em}$ and $\sqrt{D_{em}}$) due to constraints on the $E_i(p)$ term.

There are many architectural differences between the tested DNNs and biological systems. Future work may investigate how biologically plausible constraints [9] (e.g., stochasticity [48, 57], divisive normalization [58], power-law spectra [59]) and diverse training objectives [44, 13]) lead to improved neural prediction error through changes in error mode geometry.

## Acknowledgments and Disclosure of Funding

We would like to thank Nga Yu Lo, Eero Simoncelli, N Apurva Ratan Murty, Josh McDermott and Greta Tuckute for helpful discussions, and Jim DiCarlo for comments on an early version of this work. We would also like to thank the reviewers for their helpful comments, which strengthened the paper. This work was funded by the Center for Computational Neuroscience at the Flatiron Institute of the Simons Foundation and the Klingenstein-Simons Award to S.C. All experiments were performed on the Flatiron Institute high-performance computing cluster.

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

## Broader Impacts

Our work provides a novel approach towards comparing the fidelity of neural encoding model predictions, and may provide insight that leads to improved models of neural activity. Such models are applicable for neuroengineering purposes, such as retinal prosthetics or cochlear implants. However, training the deep learning models tested in the paper consumes a large amount of energy, contributing to carbon emissions. Although we used pre-trained neural networks here, future work may need to train networks with biases specifically for low neural prediction error.

## SI.1   Spectral Theory of Generalization Error in Regression

The spectral theory of generalization error for regression, as developed in [32, 33], relates the spectral properties of the data and target to the generalization error in an analytical way. Here, we summarize the key results from these works and describe how we use the results in our context.

For a generic regression task, we begin by considering a *model* $\mathbf{x}(\mathbf{s}) : \mathbb{R}^D \to \mathbb{R}^M$, which maps $D$-dimensional stimuli $\mathbf{s}$ to a set of $M$-dimensional features $\mathbf{x}$. Additionally, we assume that the *neural responses* are generated by a ground truth function $\mathbf{y}(\mathbf{s}) : \mathbb{R}^D \to \mathbb{R}^N$, mapping the stimuli to the activations of a set of $N$ biological features $\mathbf{y}$ (e.g., firing rates). Finally, we assume that the stimuli are drawn from a generic distribution $\mathbf{s} \sim \nu(\mathbf{s})$.

In regression, the goal is to estimate the ground truth $\mathbf{y}(\mathbf{s})$ from a training set of size $p$ and test it on all possible stimuli as drawn from $\nu(\mathbf{s})$. Given a training dataset with $p$-samples $\mathcal{D}_{\mathrm{tr}} = \{\mathbf{s}^\mu, \mathbf{y}^\mu\}_{\mu=1}^p$ where $\mathbf{s}^\mu \sim \nu(\mathbf{s})$ and $\mathbf{y}^\mu = \mathbf{y}(\mathbf{s}^\mu)$, ridge regression returns an estimator $\hat{\mathbf{y}}(\mathbf{s})$ as:

$$\hat{\mathbf{y}}(\mathbf{s}; \mathcal{D}_{\mathrm{tr}}) = \mathbf{x}(\mathbf{s})^\top \hat{\beta}(\mathcal{D}_{\mathrm{tr}})$$

$$\hat{\beta}(\mathcal{D}_{\mathrm{tr}}) = \underset{\beta \in \mathbb{R}^{M \times N}}{\arg\min} \sum_{\mu=1}^p ||\mathbf{y}^\mu - \mathbf{x}(\mathbf{s}^\mu)^\top \beta||^2 + \alpha_{\mathrm{reg}} ||\beta||_F^2. \tag{SI.1.1}$$

where $\alpha_{\mathrm{reg}}$ is the ridge parameter introduced to regularize the solution. Here, the regression weights and estimator depend on the particular choice of the training set $\mathcal{D}_{\mathrm{tr}}$ of size $p$. We define the normalized *generalization error* as:

$$E_g(\mathcal{D}_{\mathrm{tr}}) = \frac{1}{\int d\mathbf{s}\,\nu(\mathbf{s}) \|\mathbf{y}(\mathbf{s})\|^2} \int d\mathbf{s}\,\nu(\mathbf{s}) \|\mathbf{y}(\mathbf{s}) - \hat{\mathbf{y}}(\mathbf{s}; \mathcal{D}_{\mathrm{tr}})\|^2, \tag{SI.1.2}$$

which also depends on the training set $\mathcal{D}_{\mathrm{tr}}$. Explicit solution of the objective in Eq. SI.1 gives:

$$\hat{\mathbf{y}}(\mathbf{s}; \mathcal{D}_{\mathrm{tr}}) = \mathbf{x}(\mathbf{s})^\top \mathbf{X}_{\mathbf{1:p}}^\top \left( \mathbf{X}_{\mathbf{1:p}} \mathbf{X}_{\mathbf{1:p}}^\top + \alpha_{\mathrm{reg}} \mathbf{I} \right)^{-1} \mathbf{Y}_{\mathbf{1:p}}, \tag{SI.1.3}$$

where $\mathbf{X}_{\mathbf{1:p}} \in \mathbb{R}^{p \times M}$ is the matrix of the model responses to the stimuli in $\mathcal{D}_{tr}$, $\mathbf{Y}_{\mathbf{1:p}} \in \mathbb{R}^{p \times N}$ is the matrix of the neural responses in $\mathcal{D}_{\mathrm{tr}}$, and $\mathbf{I}$ denotes the $p \times p$ identity matrix. Defining the *kernel function* $K(\mathbf{s}, \mathbf{s}') = \mathbf{x}(\mathbf{s})^\top \mathbf{x}(\mathbf{s}')$, the predictor can be written as:

$$\hat{\mathbf{y}}(\mathbf{s}) = \mathbf{k}(\mathbf{s})^\top \left( \mathbf{K} + \alpha_{\mathrm{reg}} \mathbf{I} \right)^{-1} \mathbf{Y}_{\mathbf{1:p}}, \tag{SI.1.4}$$

where $\mathbf{K}$ is $p \times p$ matrix with elements $\mathbf{K}_{\mu\nu} \equiv K(\mathbf{s}^\mu, \mathbf{s}^\nu)$ and $\mathbf{k}(\mathbf{s})$ is a $p$-dimensional vector with elements $\mathbf{k}(\mathbf{s})_\mu \equiv K(\mathbf{s}, \mathbf{s}^\mu)$. Hence, the generalization error depends on the kernel $K(\mathbf{s}, \mathbf{s}')$, a particular draw of a training set $\mathcal{D}_{\mathrm{tr}}$ of size $p$, and the ridge parameter $\alpha_{\mathrm{reg}}$.

Note that every time a new training set is drawn from the stimuli distribution $\nu(\mathbf{s})$, the generalization error may change. We are interested in the *dataset averaged* generalization error which only depends on the size of the training set $p$ rather than the particular instances of each $D_{\mathrm{tr}}$. This is defined as:

$$E_g(p) = \left\langle E_g(\mathcal{D}_{\mathrm{tr}}) \right\rangle_{\mathcal{D}_{\mathrm{tr}} \sim \nu(\mathbf{s})} \tag{SI.1.5}$$

where the average is taken over all possible choices for $\mathcal{D}_{\mathrm{tr}}$ of size $p$. Now, this quantity only depends on the training set size $p$, input distribution $\nu(\mathbf{s})$, model kernel $K(\mathbf{s}, \mathbf{s}')$ and the ground truth function $\mathbf{y}(\mathbf{s})$. This quantity can also be thought of as the cross-validated generalization error, where the generalization errors for each regressor corresponding to a particular instance of $\mathcal{D}_{\mathrm{tr}}$ are averaged.

The theory of generalization error in [32, 33] finds that this dataset averaged $E_g(p)$ depends on the spectral properties of the kernel. Mercer's theorem [60] suggests that the kernel can be decomposed into orthogonal modes as $K(\mathbf{s}, \mathbf{s}') = \sum_i \lambda_i v_i(\mathbf{s}) v_i(\mathbf{s}')$ where the eigenvalues $\lambda_i$ and orthonormal eigenfunctions $v_i(\mathbf{s})$ are obtained by solving the equation

$$\int d\mathbf{s}' \, \nu(\mathbf{s}') K(\mathbf{s}, \mathbf{s}') v_i(\mathbf{s}') = \lambda_i v_i(\mathbf{s}), \quad \int d\mathbf{s} \, \nu(\mathbf{s}) v_i(\mathbf{s}) v_j(\mathbf{s}) = \delta_{ij}. \tag{SI.1.6}$$

Here we emphasize that the resulting eigenvalues and eigenfunctions strongly depend on the distribution of the stimuli $\nu(\mathbf{s})$. For example, for the same model kernel, the spectrum might look completely different depending on the stimuli being textures or natural scenes.

The dependence of the neural response function $\mathbf{y}(\mathbf{s})$ to the generalization error comes from the mode decomposition of $\mathbf{y}(\mathbf{s})$ on to the eigenfunction $v_i(\mathbf{s})$:

$$\mathbf{y}(\mathbf{s}) = \sum_i \mathbf{w}_i v_i(\mathbf{s}), \quad \mathbf{w}_i \equiv \int d\mathbf{s} \, \nu(\mathbf{s}) \mathbf{y}(\mathbf{s}) v_i(\mathbf{s}), \tag{SI.1.7}$$

where $\mathbf{w}_i$ is the projection of the neural response function $\mathbf{y}(\mathbf{s})$ onto the $i^{th}$ eigenfunction $v_i(\mathbf{s})$.

As described in [33, 32, 61, 62], using techniques from statistical mechanics, the generalization error is given by the self-consistent equations

$$E_g(p) = \frac{\kappa^2}{1 - \gamma} \sum_i \frac{W_i}{(p\lambda_i + \kappa)^2}, \tag{SI.1.8}$$

$$\kappa = \alpha_{\mathrm{reg}} + \kappa \sum_i \frac{\lambda_i}{p\lambda_i + \kappa}, \quad \gamma = \sum_i \frac{p\lambda_i^2}{(p\lambda_i + \kappa)^2},$$

where $\kappa$ must be solved self-consistently, and we defined the *alignment coefficients* $W_i$

$$W_i \equiv \frac{\|\mathbf{w}_i\|^2}{\int d\mathbf{s} \, \nu(\mathbf{s}) \|\mathbf{y}(\mathbf{s})\|^2} = \frac{\|\mathbf{w}_i\|^2}{\sum_j \|\mathbf{w}_j\|^2}, \tag{SI.1.9}$$

which give the ratio of the variance of neural responses along mode $i$ to its total variance. In most cases, it is not possible to find an analytical solution to $\kappa$, but it is straightforward to solve it numerically.

In this paper, since our data are finite, we take the distribution $\nu(\mathbf{s})$ to be the empirical distribution on the *full dataset* $\mathcal{D}_{\mathrm{full}} = \{\mathbf{s}^\mu, \mathbf{y}^\mu\}_{\mu=1}^P$ of $P$ data points:

$$\nu(\mathbf{s}) = \frac{1}{P} \sum_{\mu=1}^P \delta(\mathbf{s} - \mathbf{s}^\mu), \tag{SI.1.10}$$

which is an approximation to the *true* stimuli distribution where $P \to \infty$.

In this case, the functions $\mathbf{x}(\mathbf{s})$ and $\mathbf{y}(\mathbf{s})$ are fully described by the data matrices, $\mathbf{X}$ and $\mathbf{Y}$, and the generalization error in Eq. SI.1.2 becomes the neural prediction error,

$$E_g(\mathcal{D}_{\mathrm{tr}}) = \|\hat{\mathbf{Y}}(\mathcal{D}_{\mathrm{tr}}) - \mathbf{Y}\|_F^2 / \|\mathbf{Y}\|_F^2.$$

Since the full distribution of stimuli are characterized by the empirical distribution $\nu(\mathbf{s})$, training sets are generated by randomly sampling $p$ samples from $\mathcal{D}_{\mathrm{full}}$. Furthermore, the eigenvalue problem in Eq. SI.1.6 reduces to a PCA problem for the empirical kernel $\mathbf{K} = \mathbf{X}\mathbf{X}^\top$

$$\frac{1}{P}\mathbf{K}\mathbf{v}_i = \lambda_i \mathbf{v}_i, \tag{SI.1.11}$$

and the projections of neural responses $\mathbf{Y}$ on the eigenvectors $\mathbf{v}_i$ reduces to

$$\mathbf{w}_i = \mathbf{Y}^\top \mathbf{v}_i. \tag{SI.1.12}$$

We note that this method may misestimate the neural prediction error, since the true eigenvalues and eigenvectors of the kernel $K(\mathbf{s}, \mathbf{s}')$ are approximated by the finite empirical kernel matrix $\mathbf{K}$ for $P$ samples [63].

With the spectral data $\{\lambda_i\}$ and $\{\mathbf{w}_i\}$, we obtain the dataset averaged neural prediction error $E_g(p)$ (Eq. 3) as:

$$E_g(p) = \sum_i W_i E_i(p), \quad E_i(p) = \frac{\kappa^2}{1-\gamma} \frac{1}{(p\lambda_i + \kappa)^2}, \quad W_i = \|\mathbf{w}_i\|^2 / \|\mathbf{Y}\|_F^2. \quad \text{(SI.1.13)}$$

Here, we normalized the weights so that $\sum_i W_i = 1$ and $E_g(p=0) = 1$. The functions $E_i(p)$ depend only on the eigenvalues and training set size $p$. Since each $W_i$ is the normalized variance of $\mathbf{Y}$ along the eigenvector $\mathbf{v}_i$, $W_i E_i(p)$ gives the contribution to the generalization error along mode $i$ when the training set has size $p$. In our work, we defined these as *error modes* $\widetilde{W}_i(p) = W_i E_i(p)$.

To understand the behavior of the functions $E_i(p)$, we look at their limits. When $p = 0$, $E_i(0) = 1$ for all modes and no variance is explained. As $p \to \infty$, we either learn the variance $W_i$ entirely $(E_i(p \to \infty) \to 0)$ if the corresponding $\lambda_i > 0$, or we do not learn it at all if $\lambda_i = 0$ $(E_i(p \to \infty) \to 1)$. It can be further shown that, for all $p > 0$, $E_i(p) < E_j(p)$ if $\lambda_i > \lambda_j$ [33], which implies that the percent reduction in $\widetilde{W}_i(p)$ is greater for modes with larger eigenvalues.

Several remarks are in order:

- The ordered eigenvalues $\{\lambda_i\}$ characterize the *spectral bias* of the model: for a fixed training set size $p$, the reduction in $E_i(p)$ is larger for modes associated with large eigenvalues $\lambda_i$ than for modes associated with small $\lambda_i$. Therefore, a sharply decaying eigenvalue spectrum implies that there is a bias toward certain modes being learned at higher fidelity than others, while a flat eigenvalue spectrum implies that learning will be equally distributed across all modes.

- $W_i$ characterizes how much of the variance of the neural responses $\mathbf{Y}$ lie in the $i^{th}$ mode. These alignment coefficients $\{W_i\}$ characterize *task-model alignment*; because of *spectral bias*, if most of the neural response variance is associated with eigenvectors with high $\lambda_i$ terms, the neural prediction error will decrease faster compared to when most neural response variance is on eigenvectors with low $\lambda_i$. For example, if $W_i$ are largest for the modes for which $\lambda_i$ are smallest, we would expect a large generalization error when the training set size is small.

- We consider both $\{\lambda_i\}$ and $\{W_i\}$ as spectral properties of the model for the measured data (e.g. see Fig. 1C), and generalization error is determined by the interplay between these two spectral properties, together with the training set size $p$. For a fixed set of neural responses $\mathcal{D} = \{\mathbf{s}^\mu, \mathbf{y}^\mu\}_{\mu=1}^P$, different DNN models have different $\{\lambda_i\}$ and $\{W_i\}$ since for each model both eigenvalues (spectral bias) and the eigenvectors (task-model alignment) of the model activations change in a complicated way. Since we must specify both of these aspects of the model, comparison of model activations to neural responses should require at least two measures to be complete. Drawing from this argument, we may also conclude that reducing model/data comparisons to commonly used single scalar measures might be misleading as they may miss some of the factors contributing to the predictivity the others capture [64].

- In this work, we considered two geometrized measures $R_{em}$ and $\sqrt{D_{em}}$ that directly relate to the neural prediction error. The two measures jointly summarize whether the distribution of the error modes, $\widetilde{W}_i$, are relatively spread out (higher $D_{em}$, lower $R_{em}$) or tightly concentrated (lower $D_{em}$, higher $R_{em}$) at a fixed generalization error. We found that these terms were a convenient way to summarize spectral properties of the generalization error, but in principle, other measures could be defined.

## SI.2 Experimental Details

Code for analyzing the empirical and theoretical neural prediction error can be found at https://github.com/chung-neuroai-lab/SNAP.

We conducted our experiments with various vision models trained with different learning paradigms and training schemes. In total, we used 32 models and picked several model stage activations from each. Except for vision transformers (ViT), chosen model stages were intermediate convolutional model stage activations after passed through a ReLU non-linearity. For ViT, we choose intermediate encoder model stages. For untrained models, models were randomly initialized for each iteration

for each experiment. All experiments were conducted on single Nvidia H100 or A100 GPUs with 80GB RAM using PyTorch 2 [65] and JAX 0.4 [66] libraries. Additional details on model stage selection and model architectures can be found in the associated code. In the remainder of this section, we report the models used in this study and explain step-by-step how we obtain the model stage activations to perform regression.

### SI.2.1  Supervised Models

We investigated supervised vision models trained on the ImageNet 1000-way categorization task [67] with publicly available architectures and checkpoints obtained from the PyTorch library [65]. We included Alexnet [68], several ResNets [69], WideResNets [70], DenseNets [71], ConvNeXts [72], EfficientNets [73], MobileNet [74] and various vision transforms (ViT) [75].

We also investigated ANN models that were "biologically inspired" with a CorNet architecture [76]. For these models, the pre-trained weights are pulled using the library https://github.com/dicarlolab/CORnet.

In Table SI.2.1, we list all standard supervised models used in this study, together with their Top-1 ImageNet accuracies.

| Supervised | ViT | Biologically Inspired |
|---|---|---|
| alexnet - Acc: 54.92 | vit_b_32 - Acc: 75.16 | cornet_s - Acc: 72.51 |
| resnet18 - Acc: 68.98 | vit_l_16 - Acc: 85.16 | cornet_r - Acc: 53.92 |
| resnet50 - Acc: 80.63 | vit_l_32 - Acc: 76.35 | cornet_z - Acc: 45.67 |
| resnet101 - Acc: 81.71 | vit_h_14 - Acc: 85.74 | cornet_rt - Acc: 53.72 |
| resnet152 - Acc: 82.25 | | |
| wide_resnet50_2 - Acc: 78.34 | | |
| wide_resnet101_2 - Acc: 78.77 | | |
| densenet121 - Acc: 73.71 | | |
| densenet161 - Acc: 76.70 | | |
| densenet169 - Acc: 75.31 | | |
| densenet201 - Acc: 76.20 | | |
| convnext_small - Acc: 83.69 | | |
| convnext_base - Acc: 83.92 | | |
| convnext_large - Acc: 84.45 | | |
| efficientnet_b0 - Acc: 76.87 | | |
| efficientnet_b4 - Acc: 73.89 | | |
| mobilenet_v2 - Acc: 71.13 | | |

Table 1: List of supervised models studied in this work together with their Top-1 ImageNet accuracies on the validation set.

### SI.2.2  Self-Supervised and Adversarially Trained Models

We also considered models trained either with self-supervised training (SSL) [77, 78, 79] or adversarial training (referred to as "robust" models) [80, 81]. All analyzed models were ResNet50 or ResNet101 architectures.

For SSL models, we considered Barlowtwins [77], SimCLR [78] and MoCo [79] models whose pre-trained weights were pulled from https://github.com/facebookresearch/barlowtwins, https://github.com/facebookresearch/vissl and https://github.com/facebookresearch/moco-v3, respectively.

For the robust models, we obtained the pre-trained weights from https://github.com/MadryLab/robustness [50].

In Table SI.2.2, we list all SSL and robust models used in this study, together with their Top-1 ImageNet accuracies.

| SSL | Robust |
| --- | --- |
| barlowtwins - Acc: 71.68 | robust_resnet50_l2_3 - Acc: 56.25 |
| simclr_resnet50w1 - Acc: 65.71 | robust_resnet50_linf_4 - Acc: 61.26 |
| simclr_resnet101 - Acc: 69.37 | robust_resnet50_linf_8 - Acc: 46.33 |
| moco_resnet50 - Acc: 74.15 | |

Table 2: List of self-supervised (SSL) and adversarially-trained (robust) networks studied in this work, together with their Top-1 ImageNet accuracies on the validation set.

### SI.2.3   Obtaining Model Stage Activations

We obtained activations from each model stage on the stimuli from neural datasets. We employ the image preprocessing steps that were used during model training. For all models, this preprocessing is resizing image stimuli to $(224, 224)$ pixels and normalizing them with the mean and variance of the ImageNet dataset, which are $\mu = (0.485, 0.456, 0.406)$ and $\sigma = (0.229, 0.224, 0.225)$, respectively. All activations were flattened before regression.

### SI.2.4   Ridge Regression Experiment

After obtaining the model stage activations, we perform ridge regression by computing the predictions as:

$$\mathbf{Y}^*(p) = \frac{1}{p}\mathbf{X}\mathbf{X}_{\mathbf{1:p}}^\top \left(\frac{1}{p}\mathbf{X}_{\mathbf{1:p}}\mathbf{X}_{\mathbf{1:p}}^\top + \alpha_{\text{reg}}\mathbf{I}_p\right)^{-1}\mathbf{Y}_{\mathbf{1:p}}, \tag{SI.2.1}$$

where $\mathbf{I}_p$ denotes the $p \times p$ identity matrix and $\alpha_{\text{reg}}$ denotes the ridge parameter. In this notation, $\mathbf{Y}^*(p)$ is the $P \times N$ matrix of the predictions for the entire neural response set that is inferred using only a subset of $p$ samples as the training set. Then, $\mathbf{X}_{\mathbf{1:p}}$ denotes the $p \times M$ matrix of the training set that is the model activations for randomly sampled $p$ stimuli and $\mathbf{Y}_{\mathbf{1:p}}$ denotes the $p \times N$ matrix of the corresponding neural activations.

Note that the least-squares regression is ill-defined when the number of data points are more than the number of regression variables (i.e. $p > M$) since the matrix $\mathbf{X}_{1:p}\mathbf{X}_{1:p}^\top + \alpha_{\text{reg}}\mathbf{I}$ is singular for $\alpha_{\text{reg}} = 0$. In order to account for this issue, one can use the push-through identity to obtain:

$$\mathbf{Y}^*(p) = \frac{1}{p}\mathbf{X}\left(\frac{1}{p}\mathbf{X}_{\mathbf{1:p}}^\top\mathbf{X}_{\mathbf{1:p}} + \alpha_{\text{reg}}\mathbf{I}_M\right)^{-1}\mathbf{X}_{\mathbf{1:p}}^\top\mathbf{Y}_{\mathbf{1:p}}, \tag{SI.2.2}$$

where now $\mathbf{I}_M$ is the $M \times M$ identity matrix. The form in Eq. SI.2.2 is the conventional form of ridge regression, while the equation Eq. SI.2.1 is the form obtained via the so-called *kernel trick* [82, 36]. However, we only use the form in Eq. SI.2.1, since the model activations often are high-dimensional ($\sim 1M$) and hence $M \gg P$, where $P \sim 10^3$ or less.

We repeat the same computation over 5 random choices of a training set and report neural prediction error Eq. SI.1.11 averaged over these trials. Furthermore, in Sec. SI.4, we show the agreement between empirically calculated and theoretically predicted errors.

In our experiments, we choose $\alpha_{\text{reg}} = 10^{-14}$ for numerical stability, which roughly corresponds to double-precision. Note that this choice of ridge parameter is to get close to least-squares regression, in which case $\alpha_{\text{reg}}$ is identically zero. We considered a very small ridge parameter, since even for ridgeless regression, strong spectral biases of overparameterized models avoid overfitting and effectively behave as if there is a non-zero ridge parameter [83, 32, 40, 84, 33, 85].

Here we chose a single regularization parameter for all experiments, however the choice of regularization parameters and the relation to the error modes is critical for fully characterizing the responses of encoding models. Future work could address this by choosing ridge parameters separately for different models and model stages, or allowing flexibility for different regularizations for individual unit responses (rather than one regularization for each neural region).

### SI.2.5   Computing Theoretical Error Mode Geometry

After obtaining the model stage activations, we compute the eigenvalues and eigenvectors of each activation using the formula in Eq. SI.1.6. In terms of empirical Gram matrices, this corresponds to

performing Principal Component Analysis (PCA) on the Gram matrix. We project the neural response data onto the obtained eigenvectors and calculate $W_i$. Then, theoretical values of $\sqrt{D_{em}}$ and $R_{em}$ are calculated using the formula given in Eq. 5. For each $p$, the value of $\kappa$ is solved self-consistently using the JAX auto-differentiation library [66] and SciPy optimizers [86].

## SI.3 Relation to Brain-Score Neural Prediction Metrics

In this work, we considered ridge regression to compare model activations to neural responses. While there are many other choices for model comparisons such as canonical correlation analysis (CCA), partial least squares regression (PLS) and kernel-target alignment (KTA) [22, 23, 51], we chose to focus on ridge regression as it is theoretically well studied in terms of spectral properties [32, 33].

To demonstrate that the neural prediction error we obtained from ridge regression is comparable to other widely-used metrics, we directly compared our predicted values to those obtained from PLS using the Brain-Score [11] framework. Brain-Score uses PLS regression and reports the noise-corrected Pearson $r^2$ score on test data to rank models. Note that PLS regression finds a new basis that maximizes the correlations between two sets. Then the regression is done on new variables by performing dimensionality reduction on both sets using the common basis. If one uses all the components in this new basis, then PLS is equivalent to least squares regression.

To calculate the PLS Pearson $r^2$ value, we first extracted model activations as described in SI.2.3. We then fit PLS regression using the standard 25 latent components [11] and calculated the Pearson correlations between PLS-based predictions and actual neural responses on held-out data. We took the median correlation across all neurons and divided it by the median noise ceiling, which was calculated using split-half reliability across trials featuring identical stimuli. Train and test data points were randomly assigned, and the final Pearson $r^2$ values were calculated by averaging over 10 random splits. This procedure is described in more detail in [11] and corresponds to the default analysis for public Brain-Score neural prediction benchmarks in the following regions and datasets:

    V1: `movshon.FreemanZiemba2013public.V1-pls`
    V2: `movshon.FreemanZiemba2013public.V2-pls`
    V4: `dicarlo.MajajHong2015public.V4-pls`
    IT: `dicarlo.MajajHong2015public.IT-pls`

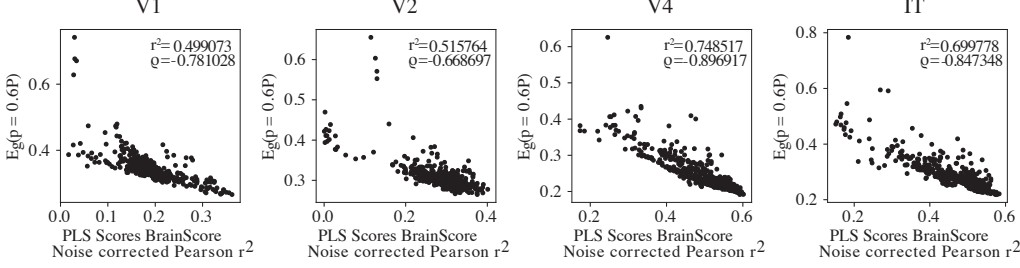

Figure SI.3.1: **Brain-Score PLS regression $r^2$ values vs. ridge regression empirical $E_g(p)$ values.** Each point corresponds to the predictions obtained for a single model and stage on the corresponding region data. Pearson $r^2$ values and Spearman $\rho$ values are given for the correlation between the Brain-Score value and the neural prediction error. In all regions, there is a general trend where models that do well on one metric also do well on the other metric. A train test split of 60%/40% is used for our ridge regression procedure and the 90%/10% split for Brain-Score PLS.

In Fig. SI.3.1, we compare the Brain-Score Pearson $r^2$ from each model stage to the empirical ridge-regression $E_g(p)$. The Brain-Score metric and neural prediction error from ridge regression are moderately correlated, despite the fact that we are (1) comparing values between ridge regression and PLS regression and (2) comparing a Pearson $r^2$ metric to $E_g(p)$ values, which are not directly comparable (Pearson $r^2$ measures the angle between predictions and true labels, while neural prediction error measures the distance between the two).

To directly address the difference in metrics, in Fig. SI.3.2, we compare the Brain-Score PLS regression Pearson $r^2$ on the test data to ridge regression Pearson $r^2$ on the test data (using the same regression weights as were used to obtain the $E_g(p)$ values above. When using the same metrics on the two different regression methods, there is a strong correlation for all regions. Combined, these results show that by using the ridge regression formulation we generally preserve the ordering of the commonly used Brain-Score metric.

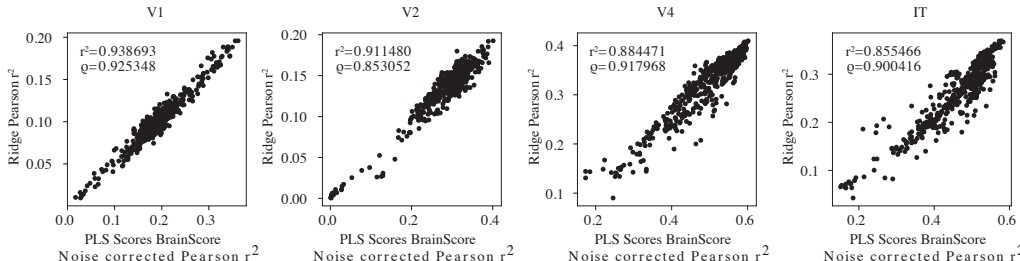

Figure SI.3.2: **Brain-Score PLS regression Pearson $r^2$ values vs. ridge regression Pearson $r^2$ values on test data.** Each point corresponds to the predictions obtained for a single model and stage on the corresponding region data. Pearson $r^2$ values and Spearman $\rho$ values are given for the correlation between the Brain-Score value and the Pearson correlation obtained from the test set in the ridge regression. Note that our ridge regression Pearson $r^2$ values are not noise corrected, but if we were to do so, it would correspond to dividing by a scalar value for each region (the noise ceiling obtained from the population). The same models and predictions are shown in Fig. SI.3.1, including the train test split of 60%/40% for our ridge regression procedure and the 90%/10% split for Brain-Score PLS.

## SI.4    Empirical vs. Theoretical Neural Prediction Error

Complimentary to the discussion around Fig. 3, we confirmed that the theoretical predictions (see Sec. SI.1) obtained from the spectral properties of data match with the empirical regression discussed in Sec. SI.2 for various values of $p$. In Fig. SI.4.1, we plot the learning curves for selected model stages of a pre-trained ResNet18 architecture for all brain regions. We see that the theory matches the empirical values for all regions.

Note that particular choices of train/test splits may alter the ranking of which encoding models have the lowest neural prediction error (for instance, see layer4.1.relu changing from the lowest neural prediction error at small $p$ to the highest error at large $p$ in Fig SI.4.1 IT). Additionally, as we mentioned in the main text, increasing the train/test split yielded worse agreement between empirical and theoretical neural prediction errors. The main reason for this disagreement is that the theoretical values depend highly on the estimation of the true eigenvalue distribution of the empirical Gram matrix of data. However, as discussed in Sec. SI.1, the tail of the true eigenvalues are often underestimated by the empirical eigenvalues and that creates a discrepancy between observed $E_g(p)$ and theoretically predicted one [63]. This discrepancy can be clearly seen in Fig. SI.4.2, where we plot the empirical neural prediction errors against the theoretically predicted ones. As the amount of training data increases (corresponding to a decrease in amount of test data), the correlation between the two quantities decreases monotonically.

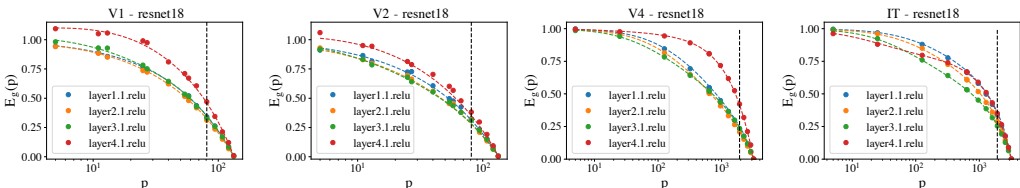

Figure SI.4.1: **Empirical vs. theoretical learning curves.** In this figure, we show the learning curves for regression from selected model stages in a trained ResNet18 architecture to all brain regions. The colored dots correspond to the mean empirical error over 5 trials, and the colored dashed lines correspond to the theoretical learning curves obtained by the formula for $E_g(p)$. The vertical black line corresponds to the 60 train / 40 test split used for the main analysis in this paper.

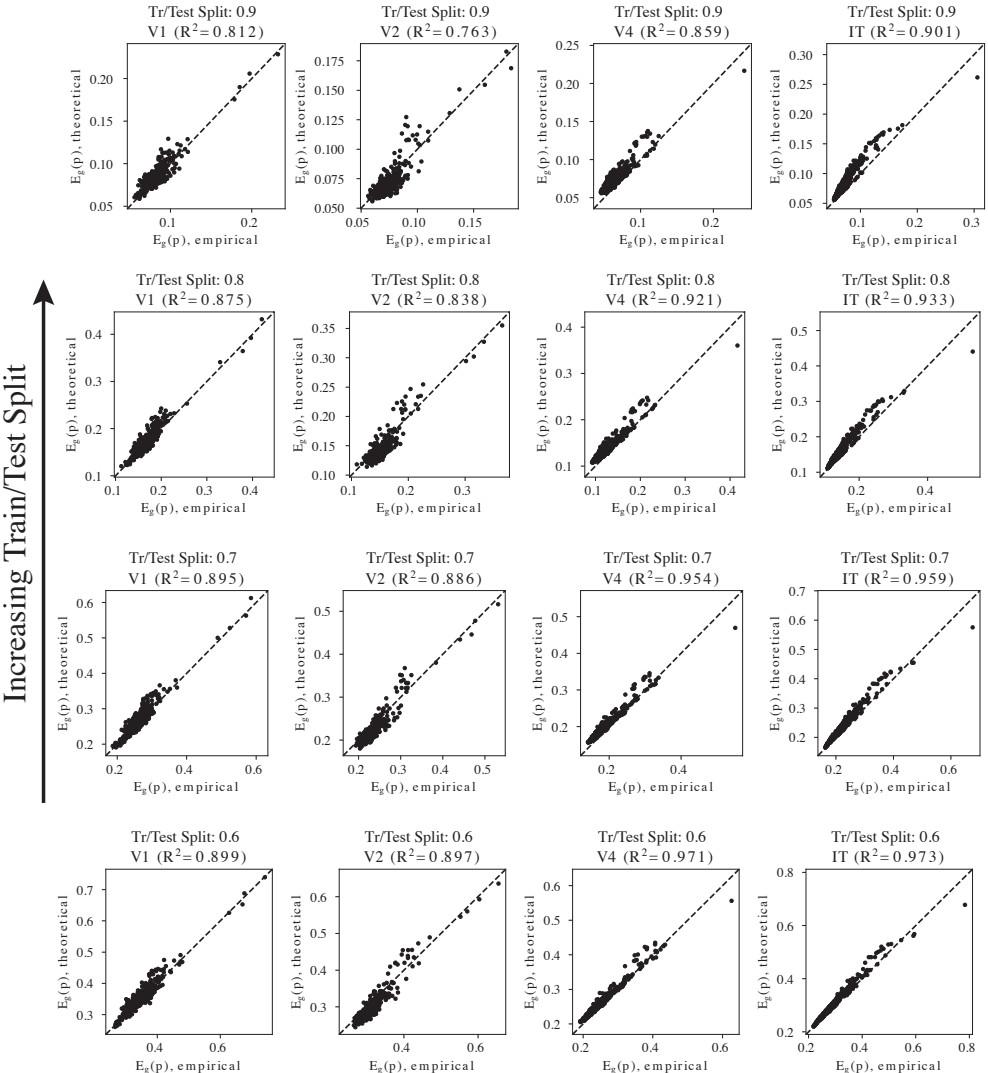

Figure SI.4.2: **Empirical and theoretical predictions worsen with train/test split.** We show the empirical values of ridge regression $E_g(p)$ compared to the theoretical values. As the amount of data used for training increases (corresponding to a decrease in test data), the correlation between empirical and theoretical values decreases.

# SI.5 Additional Experimental Results

## SI.5.1 Geometry grouped by architecture and training types

For visualization purposes, we show the error mode geometries grouped by the architecture (Fig. SI.5.1) and for a subset of models all with ResNet50 backbones but with different training types (Fig. SI.5.2).

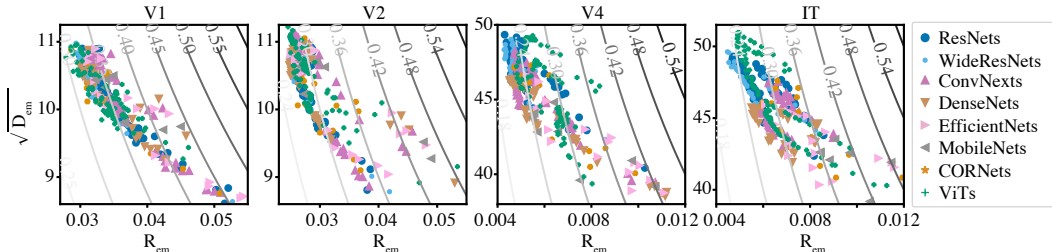

Figure SI.5.1: **Error mode geometry for supervised networks grouped by architecture type.** All models included were trained in a supervised manner on ImageNet (standard training without any adversarial perturbations) and were obtained from publicly available checkpoints in the PyTorch Library, with the exception of CORNets which were obtained from https://github.com/dicarlolab/CORNet. Model groups were ResNet architectures (`resnet18`, `resnet50`, `resnet101`, `resnet152`), Wide-ResNet architectures (`wide_resnet50_2`, `wide_resnet101_2`), ConvNext architectures (`convnext_base`, `convnext_small`, `convnext_large`), DenseNet architectures (`densenet121`, `densenet161`, `densenet169`, `densenet201`), EfficientNet architectures (`efficientnet_b0`, `efficientnet_b4`), MobileNet architectures (`mobilenet_v2`), CORNet architectures (`cornet_r`, `cornet_rt`, `cornet_s`, `cornet_z`), and ViT architectures (`vit_b_32`, `vit_h_14`, `vit_l_16`, `vit_l_32`).

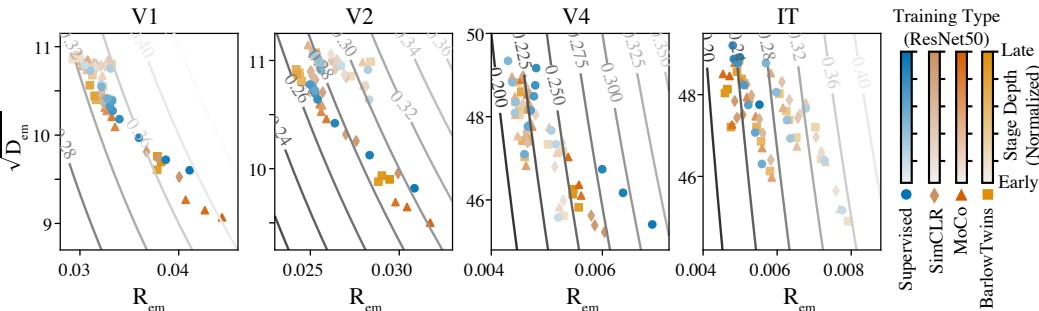

Figure SI.5.2: **Error mode geometry for supervised and self-supervised ResNet50 architectures.** Supervised ResNet50 architecture (trained on ImageNet without any adversarial perturbations) was compared to ResNet50 architectures trained with self-supervised methods SimCLR [78], MoCo [79], and BarlowTwins [77] There are no clear differences in geometries between models with different training mechanisms. This is in line with previous work that found similarities in representations between supervised and self-supervised networks [87, 88, 89].

## SI.5.2 Geometry across model stages

In the main text, we focused on data from regions V1 and IT when analyzing the error mode geometry's dependence on model stage depth. Analysis of regions V2 and V4 is included in Fig. SI.5.3. Additionally, to show the error mode geometries from different model stages in Fig. 4A and Fig. SI.5.3A, we zoomed in on the parts of the space with the most data points and maintained the same axes limits for regions that shared the same stimulus set. In Fig. SI.5.4, we present all model stages, including outliers, colored by model stage depth.

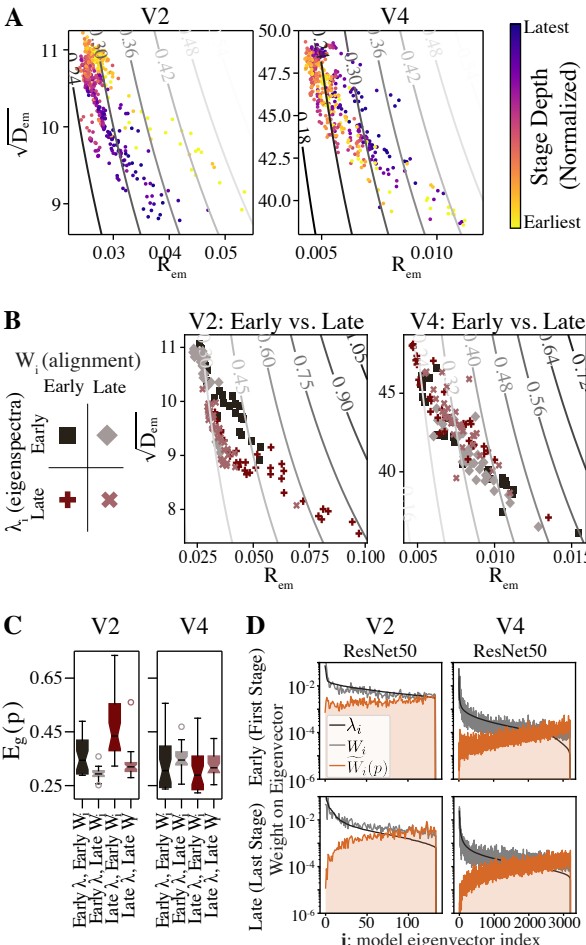

Figure SI.5.3: **Relation of error mode geometry with model stage depth in V2 and V4.** **(A)** Each point represents error mode geometry for a specific model stage of trained neural network models, where the color of the value represents the depth of the model stage. For each model, the color code is normalized in the $[0, 1]$ range where the earlier and later model stages have lighter and darker colors, respectively. $R_{em}$ and $\sqrt{D_{em}}$ values are obtained from theoretical values, and contour lines correspond to theoretical values of the neural prediction error as in Fig. 2. **(B,C)** Predicted $E_g(p)$ that would be obtained when using the eigenspectra from the first (Early) or last (Late) stage of each model, paired with the alignment $W_i$ terms from the Early or Late stage of each model. Full error mode geometry is given in (B), while summarized bar plots of the $E_g(p)$ from each comparison is given in (C). **(D)** Full spectra for $\lambda_i$, $W_i$ and $\widetilde{W}_i$ for Early and Late model stages of V2 and V4 datasets.

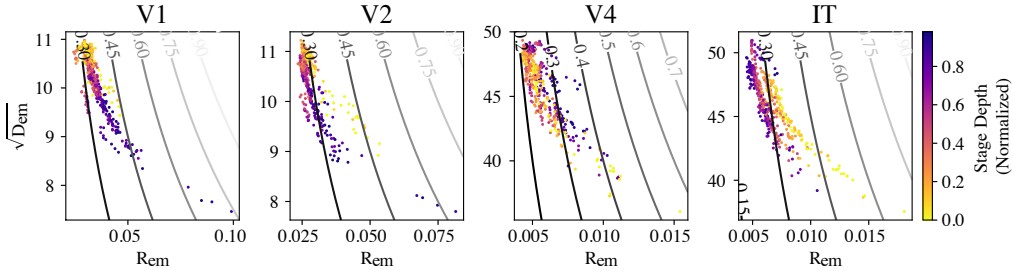

Figure SI.5.4: **Error mode geometry for trained neural networks, colored by model stage depth.** Same plots as Fig. 4A and Fig. SI.5.3A, but including the outliers.

We further elaborate on the spectral properties of different model stages and the corresponding error mode geometry by using the $\ell_\infty$ trained ResNet50 [50] model as an example. We plot the eigenvalues of the model activations from each model stage for all tested visual regions(Fig. SI.5.5). For V1 and V2, we saw that earlier model stages had better predictions, while for V4 and IT, later model stages performed better.

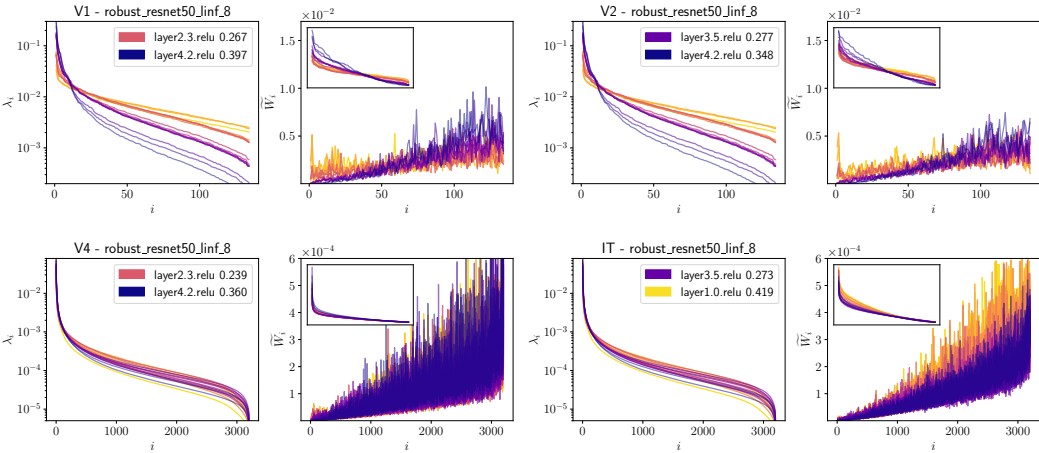

Figure SI.5.5: **Spectral properties of $\ell_\infty$ trained ResNet50 model on all regions.** For each region (V1, V2, V4, IT), the left panel shows the eigenvalues of the model stages. The legend shows the best and worst model stage in terms of neural prediction error, and the colors correspond to the colors associated with that model stage (see Fig. 4). The right panel shows the error mode spectrum for each model stage, with the inset showing the ordered spectrum.

The spectral properties of each model stage depend highly on the input distribution. We notice that, under the input distribution of V1/V2 data, the eigenvalues of earlier model stages decayed more slowly than later model stages, while under the input distribution of V4/IT data this pattern is reversed where later model stages decayed more slowly. These differences are not due to the recorded neurons, as the eigenspectra is only a function of the stimulus set, pointing out the importance of considering the stimuli used for the neural predictions. Future work could consider choosing stimulus sets to maximize the difference between model eigenspectra. This would likely lead to greater differences between models in their predictions of neural activity patterns.

## SI.5.3 Geometry for Trained vs. Untrained Models

In the main text analysis of trained vs. untrained models, we zoomed in on the parts of the space with the most data points and maintained the same axes limits for regions that shared the same stimulus set (Fig. 5)A. In Fig. SI.5.6, we show all the points of the trained and untrained networks, including the outliers. We also include an analysis of the error mode geometry for trained vs. untrained networks when predicting an additional V1 dataset [46] Fig. SI.5.7.

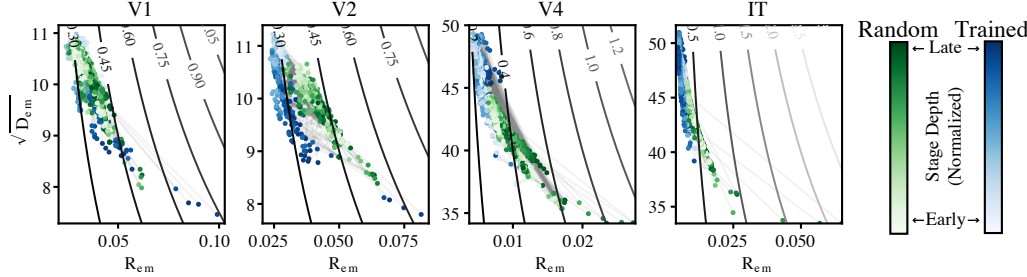

Figure SI.5.6: **Error mode geometry changes with model training, full plots showing outliers.** Same plot as Fig. 5A, but including the outliers

.

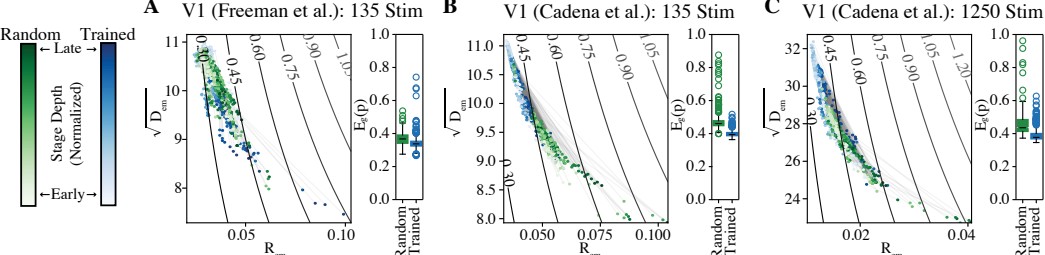

Figure SI.5.7: **Error mode geometry for V1 dataset from [46]**. Random and Trained error mode geometry for **(A)** previously analyzed V1 dataset, **(B)** V1 natural image dataset from [46] matched to stimulus set size of **(A)**, and **(C)** using full V1 natural image dataset from [46]. In all cases, there was a small decrease in $E_g(p)$ when using the trained neural network activations to predict the neural data (but this difference was less notable than in some of the other regions). Increasing the dataset size from 135 to 1250 stimuli did not change the overall trends.

## SI.5.4 Geometry for Robust vs. Standard Models

In this section, we consider the model stage-wise changes in error mode geometry for an $\ell_\infty$ adversarially trained network, as opposed to the $\ell_2$ trained network presented in Fig. 6A. In Fig. SI.5.8, we find that the error mode geometry is very similar for this network, suggesting a consistent effect of adversarial training on the error mode geometry.

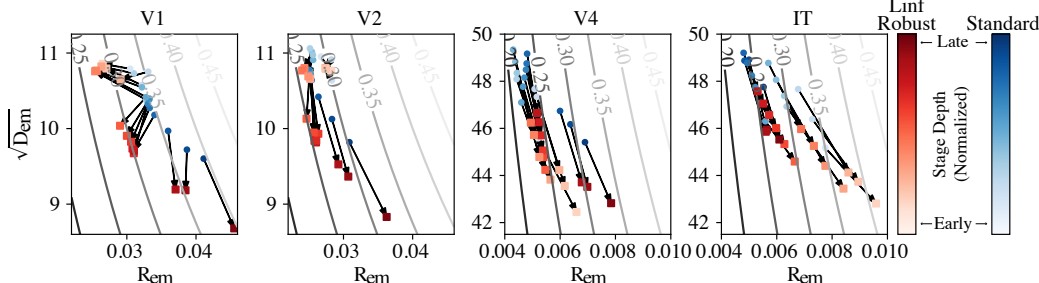

Figure SI.5.8: **Error mode geometry for $\ell_\infty$ adversarially trained model.** Same plot as Fig. 6A, but shown for an $\ell_\infty$ trained ResNet50.

For visualization purposes, in Fig. SI.5.9, we plot the eigenspectra and error mode weights for a standard and adversarially trained model.

## SI.5.5 Effective Dimensionality and Neural Prediction Error

Previous work has suggested that the effective dimensionality of a model's responses to visual stimuli correlates strongly with its neural predictivity [31]. Here, we did not find a relationship between neural prediction error and the dimensionality of the model activations. We measured the effective dimensionality ($ED$) of the model stage responses for the stimuli in each neural experiment, extracted as described in SI.2.3, with the participation ratio of the eigenvalues of $\mathbf{G_X}$:

$$ED \equiv \frac{(\sum_i \lambda_i)^2}{\sum_i \lambda_i^2}. \tag{SI.5.1}$$

We tested the correlation between $ED$ and both Brain-Score performance and neural prediction error. In Fig. SI.5.10, we show that neural prediction error does not correlate well with model dimensionality across any of the datasets considered here. This is to be expected based on the theory presented – although the spectral properties of the model activations to the stimulus set is one of the factors

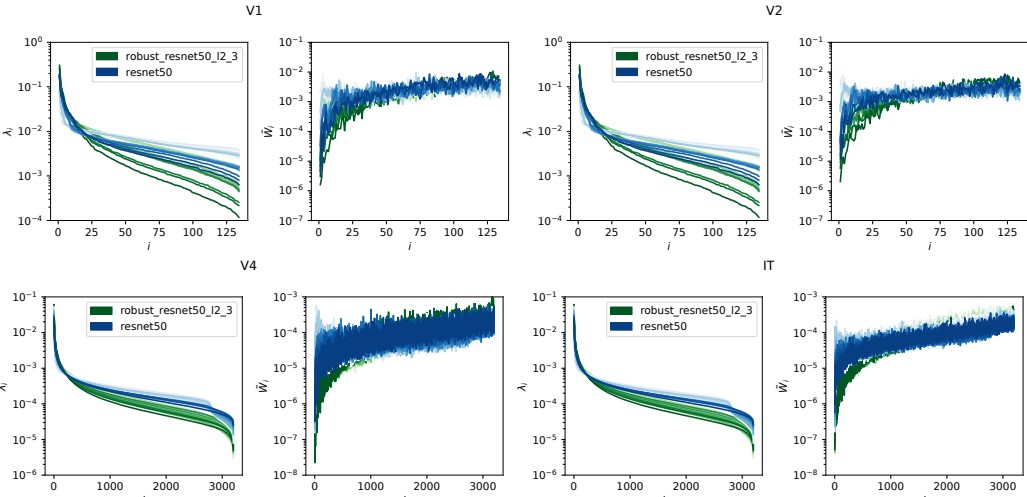

Figure SI.5.9: **Spectral plots for standard and adversarially robust ResNet50.** Eigenvalue spectra and error mode weights are given for all model stages of a standard ResNet50 (blue) and $\ell_2(\epsilon = 3)$ Robust ResNet50 (green) for all brain regions.

leading to the observed values of $\widetilde{W}_i$, the direction of the model eigenvectors is also important for determining the error mode geometry. These properties are captured by our proposed geometric quantities $R_{em}$ and $D_{em}$, which are directly related to the neural prediction error. Additionally, in the main text, we investigated the neural prediction error in cases where we modified the eigenvalues or alignment coefficients (which are dependent on the model eigenvectors).

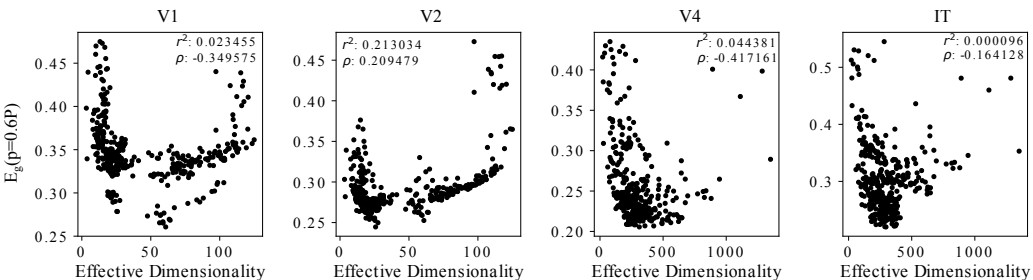

Figure SI.5.10: **Effective dimensionality from model activations vs.** $E_g(p)$ **values for each neural region.** When the effective dimensionality and generalization error are measured using the full space of model activations, we do not see a strong correlation with either Pearson correlation ($r^2$) or Spearman correlation ($\rho$) between $ED$ and $E_g(p)$.

We note that the authors of [31] applied either an average-pooling or principle-components-based dimensionality reduction method to model activations before carrying out their regression analyses. To ensure that we were able to reproduce the results of [31], we performed similar pooling on our model activations, which is complete average-pooling on activations from each convolutional filter. When average-pooling to our model activations, we found that setting $\alpha_{\text{reg}} = 10^{-14}$ as above yielded several outliers with unusually high $E_g$ values. We therefore set $\alpha_{\text{reg}}$ to the value that yielded the lowest possible $E_g$ for each set of model activations. When pooling model activations, we found a relatively strong relationship between prediction error and model dimensionality in IT similar to that reported in [31] (Fig. SI.5.11). We further replicated the results in [31] by directly comparing the effective dimensionalities obtained from the spatially averaged activations to the Brain-Score noise-corrected PLS regression values (Fig. SI.5.12). Taken together, these results indicate that pooling discards important variability in the model responses, and that when we use the full model activations the model dimensionality does not correlate well with neural prediction error.

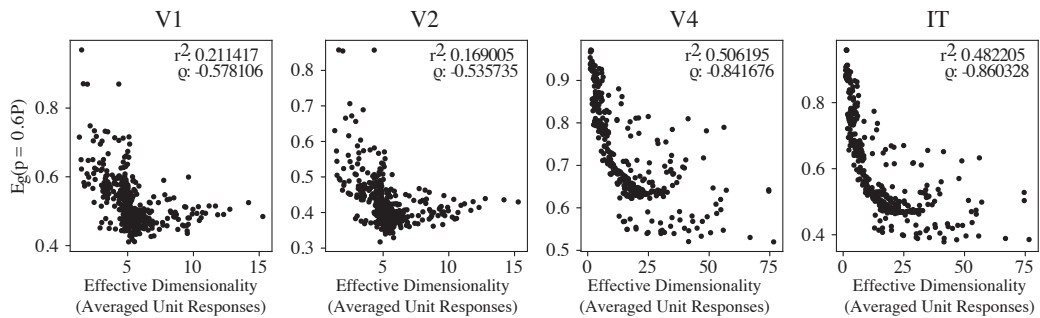

Figure SI.5.11: **Effective dimensionality from spatially averaged unit activations to stimulus set vs. $E_g(p)$ values for each neural region.** When the effective dimensionality and regression are both performed on the spatially averaged unit activations, we see a stronger correlation between the effective dimensionality and the neural responses, particularly in IT and V4. However, the neural prediction errors are generally higher when using these pooled responses (note the different y axes compared with Fig. SI.5.10), demonstrating that this type of pooling discards much of the important variability in the model responses

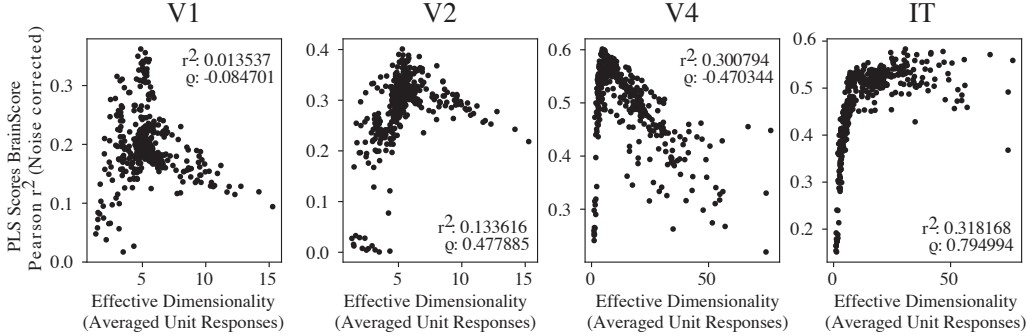

Figure SI.5.12: **Effective dimensionality from spatially averaged unit activations vs. Brain-Score values for each region.** When the effective dimensionality is calculated on the spatially averaged unit responses, rather than the full space of activations, we see a correlation that mirrors that reported in [31] for IT responses. In other brain regions, the relationship is less monotonic, appearing as the "joint regime" discussed in [31]. Note that the Brain-Score metric does not rely on taking the spatial average across units, but does perform dimensionality reduction as implemented with PLS.

