# OpenReview forum: "A Spectral Theory of Neural Prediction and Alignment"
_NeurIPS.cc/2023/Conference — NeurIPS 2023 spotlight_

### Official Review · Reviewer_EnLw · 2023-06-28

**Soundness:** 2 fair
**Presentation:** 2 fair
**Contribution:** 2 fair
**Rating:** 4
**Confidence:** 4

**Summary:**

This submission uses spectral theory and simulation experiments to try to compare and  assess the representations of DNNs vs those of biological neural networks.

**Strengths:**

The paper comes with a fairly extensive review of the literature.
The paper uses both theoretical ideas and simulations.

**Weaknesses:**

The paper is difficult to read and understand. The primary reason is that the problem being tackled is not stated in any clear way.  The boundary between artificial NNs and biological NNs is very fuzzy throughout the paper and it is often unclear if the authors are discussing biological NNs or artificial NNs. The results also are not very clear. This is already obvious from the abstract which does not contain a single quantitative result. Furthermore, there are too many concepts piled up in the paper in a messy way (spectral, geometry, alignment, adversarial vs standard training, error mode weights, etc.).

The authors cite 43 papers. If the goal is to provide a fairly complete overview of the relevant work, then it seems that some key papers are missing (e.g. Anderson and Zipser in the 1980s, more recent work by DiCarlo et al. )

**Questions:**

There may be good ideas behind this paper and in time this direction of research may prove useful. But at this stage things look premature and very foggy. In future versions, it will be essential to improve the clarity and keep very clear distinctions between artificial and biological NNs. A term like "neural predictivity" is extremely ambiguous in your context as it could have several meanings.

**Limitations:**

In addition to the problems mentioned above, there are significant limitations posed by the size of the data sets. The authors are aware of this problem and mention it in Section 5 together with other limitations.

---

> ### Author Rebuttal · Authors · 2023-08-10
>
> We thank you for raising these issues and your comments. Please see below for the responses.
>
> 1. **Clarity of the results**
>
> We have made several changes to the paper which we believe substantially address these issues. While we refer the reviewer to the global response for details, we note that we have substantially revised the figures of the paper to give clear schematics of the modeling problem that we consider and describe our results (Rebuttal Figures 1&2), streamlined the presentation of the theory in Sec. 2, and added a concise list of our main contributions to the first section.
>
>
> 2. **Additional Citations**
>
> Thank you for suggesting that we provide additional context for how our results relate to previous literature. Within this work, we are primarily discussing taking the activations from computational models (pre-trained neural network models or other hand-engineered stimulus computable models) and learning a regression mapping between the activations of the computational model and those measured in the brain for the same set of stimuli. This is directly related to the lines of work from DiCarlo et al. We already cite 6 papers from the group, but have also added the following citation to our section on Adversarial vs. Standard Training:
>
> * Guo, C., Lee, M., Leclerc, G., Dapello, J., Rao, Y., Madry, A., & Dicarlo, J. (2022, June). Adversarially trained neural representations are already as robust as biological neural representations. In International Conference on Machine Learning (pp. 8072-8081). PMLR.
>
>
>
>
> Regarding the Andersen and Zipser work, this is more along the lines of building non-linear models that predict neural responses. There are many recent works along these lines–see e.g., [1,2]. It is beyond the scope of this paper to apply our theory to this framework, but we have added the following to the discussion for future work along these lines:
>
> “Finally, we note that while we focused on DNN models trained on image data sets, a closely related line of work has investigated the properties of DNN models directly trained to predict neural responses  [1-3]. Future work can compare the error mode geometries from end-to-end networks to those obtained from networks trained on image data. Such an analysis could elucidate how these two vastly different training schemes are both able to predict neural data well. Furthermore, the spectral theory of kernel regression we used here has previously been applied to analyze kernels corresponding to different layers of DNNs at different stages of the training process [4]. Future work could take a similar approach to analyzing how representational geometry evolves throughout the training process of end-to-end DNN models by extracting layer wise kernels at different training checkpoints.”
>
> ## Questions
>
> 1. **Ambiguity in the term, "neural predictivity"**
>
> We believe that our revisions make the manuscript substantially clearer. However, we decided to continue to use the term "neural predictivity," as this is a standard term in this field. See for example [5,6] below.
>
>
> ## Limitations
>
> 1. **Data set size.**
>
> In response to reviewer concerns regarding the data set size for V1, we reran these experiments on a significantly larger data set from Ref. [7] with $P=1250$ images (see response to Reviewer t8VM and Rebuttal Figure 6). Note that the V4 and IT data sets are significantly larger and are standard data sets for bench marking models of visual cortex [8]. Additionally, note that the V1/V2 data sets that we used were a small, publicly available, subset of the full data presented in [9]. We are in the process of obtaining the full data.
>
> References
>
> [1] Cadena et. al., PLoS Comp. Bio., 2019
>
> [2] Klindt et. al., NeurIPS, 2017
>
> [3] Zipser & Andersen, Nature, 1988
>
> [4] Canatar & Pehlevan, IEEE, 2022
>
> [5] Zhuang et. al., PNAS, 2021
>
> [6] Yamins et. al., PNAS, 2014
>
> [7] Cadena et. al., PLoS Comp. Bio., 2019
>
> [8] Shrimpf et. al., BioRxiv, 2018
>
> [9] Freeman et. al., Nature Neuro, 2013

---

> > ### Comment · Reviewer_EnLw · 2023-08-20
> >
> > Thank you for the comments and clarification. I will increase my score by 1.

---

### Official Review · Reviewer_t8VM · 2023-07-04

**Soundness:** 3 good
**Presentation:** 2 fair
**Contribution:** 3 good
**Rating:** 7
**Confidence:** 3

**Summary:**

The paper uses (but doesn't introduce) a theoretical framework that relates generalization error to spectral bias of network activations to geometrically/spectrally analyze what aspects of pretrained deep networks contribute to the predictive performance of neural activity in layers V1, V2, V4 and IT. The authors verify that the theory matches empirical values, and analyze different aspects of the generalization, such as the influence of the training data size and adversarial pretraining.

**Strengths:**

It is inherently difficult to analyze what aspects of deep network representations contribute to the predictive performance on neural data. The authors make an important contribution that disentangle different factors. This will help to understand why and when deep representations predict neural data well.

**Weaknesses:**

My main two criticisms are:
- The clarity/accessibility could be improved at times.
- There are a few more references that could be considered.
- Some of the assumptions/limitations could be discussed more clearly.

While I like the general approach, I think it could be clearer at some point. Since you, as the authors, have thought a long time about what the different $W_i$, $R$, $D$ and other values mean, the reader sees that for the first time. So I think it's important to give a good intuition about what they mean. Figure 1 didn't really help in that respect. Particularly, D and E need a bit more effort to be helpful. The reason why this is so crucial is that they mainly discuss the observations in terms of these values. So if the reader doesn't have a clear intuition what they mean, the observations/interpretations become meaningless. So I urge the authors to improve on the introduction part and on the observation part (--> Q1).

I think the following papers would be worth considering to cite (no, I am not S. Cadena ;) ):
* Santiago A. Cadena, Konstantin F. Willeke, Kelli Restivo, George Denfield, Fabian H. Sinz, Matthias Bethge, Andreas S. Tolias, Alexander S. Ecker Diverse task-driven modeling of macaque V4 reveals functional specialization towards semantic tasks
* Santiago A. Cadena, Fabian H. Sinz, Taliah Muhammad, Emmanouil Froudarakis, Erick Cobos, Edgar Y. Walker, Jake Reimer, Matthias Bethge, Andreas Tolias, Alexander S. Ecker How well do deep neural networks trained on object recognition characterize the mouse visual system?
* S. A. Cadena, G. H. Denfield, E. Y. Walker, L. A. Gatys, A. S. Tolias, M. Bethge, and A. S. Ecker Deep convolutional models improve predictions of macaque V1 responses to natural images

The first, because it discusses different prediction performances depending on the pretrained representation. The second because it finds that random representations do almost equally well as pretrained (albeit for mouse). And the third, because it compared task driven and data driven prediction performance for monkey V1. As far as I know, the authors also provide the data. So you could even repeat some of the analysis with more data for V1 (135 texture stimuli are not a lot).

Re Limitations/Assumptions: Some of the assumptions seem unrealistic to me. I would ask the authors to discuss that in more detail. Specifically
- Q2: You write that model and brain responses need to be deterministic functions of the stimuli. For brain responses this is certainly not true since there is noise and also latent brain states. Can you discuss how the deviation from this assumption affects your results.
- Q3: I am unsure about the  $M,P\rightarrow \infty$ and $M/P \in O(1)$ assumption. This means that the model features need to go to infinity as the data grows. Doesn't that mean that the model has to be non-parametric, which is not true for deep network. I would ask the authors to discuss that in more detail.
Discussing the limitations in two very short paragraphs in section 5 is not enough. Especially, since they have two pages left.

**Questions:**

Q1-Q3: See above.
Q4: I couldn't find how you extracted the 516 model activations (neither in the paper nor in the supp material). Please briefly describe this procedure in the main paper (i.e. did you use PCA? on how many activations/stimuli? did you only use one layer or many?).

Minor:
- l 97: "self-consistently" did you mean "self-consistency"?

**Limitations:**

See weakness above.

---

> ### Author Rebuttal · Authors · 2023-08-10
>
> We thank you for your thorough review. Please see our responses below.
>
> **Q1:**
> Thank you for the suggestions regarding the clarity and presentation of the manuscript. We have made a number of changes that we hope address these concerns. These are detailed in the global response. We have also updated Fig. 1 (see RFig 1) and have expanded on Fig. 1E into its own Figure which we hope will guide interpretation of our results (RFig 2).
>
>
> **Re Cadena References:** Thank you for the references. All of these are very related to our work. We have added the following references:
>
>
> 1. *Diverse task-driven modeling of macaque V4 reveals functional specialization towards semantic tasks:* We reference this work while discussing how training typically benefits models in later visual regions more than earlier ones (see global response).
> 2. *How well do deep neural networks trained on object recognition characterize the mouse visual system?:* We now reference this work in the revised section presenting our results regarding trained vs. random networks (see global response).
>
>
> For the third reference, we analyzed the trained vs. random network error mode geometry using this V1 dataset (focusing on the $P=1250$ natural images, RFig 6), as suggested. This analysis reproduced the overall trends where trained networks do slightly better than random networks at predicting V1 neural data, but not by much. This finding was consistent when using 135 images (matched to the previous V1 dataset) or when using all 1250 images. Additionally, the same trends held when using neural recordings from Cadena et al. texture stimuli rather than the natural images (not shown for space reasons). Even though the overall trends replicated, we found that the generalization error for the Cadena et al. dataset was higher (corresponding to worse predictions) than observed in the Freeman et al. dataset. We are further investigating this difference, but it may stem from a lower number of trial presentations per stimulus in the Cadena et al. dataset (2 or 4 trials per stimulus, in comparison to 20 presentations for the Freeman et al. dataset) leading to larger noise levels. We will include RFig 6 in the SI of the paper.
>
> **Q2:**
> Thank you for this comment. Previous works on the spectral theory of ridge regression have investigated the role that noise plays in this setting [1]. In the kernel regression setting, noise leads to an additional term in the generalization error formula which depends on the spectrum of the model. Explicitly modeling neuronal noise is an important direction for future work, and we expanded the following paragraph in the limitations section:
>
> “*...Moreover, this theory assumed that the brain responses are deterministic functions of the stimuli. As shown in [1], this assumption may be relaxed, and future work can examine the role of neuronal noise in these contexts.*” (Quote follows line 259 in the original manuscript.)
>
> **Q3:**
>
> This limit is often employed to obtain asymptotic expressions whenever both the number of units and the number of data points is large–see for example [1-3]. Our framework does not require that the feature dimension $M$ grows with $P$. Here, we focus on this large $M$ and $P$ limit since the resulting theory accurately predicts the performance of kernel ridge regression when applied to real data. However, since our formula for $E_g$ is only exactly correct in the large $M,P$ limit, we explicitly confirmed that this theory predicted empirical generalization errors in Fig. 2A.
>
> **Q4:**
>     Thank you for highlighting this confusion. We do not apply any dimensionality reduction to the model activations and flatten activations from convolutional layers. As it was shared by other reviewers we have included more details of the architecture, layer choices, and extraction on lines 76-80 of the main text:
> “*The neural responses and model activations were extracted for each stimulus (Fig. 1B), and each layer of each investigated neural network was treated as a separate encoding model. We examined 32 deep neural networks trained on the ImageNet classification task. Model architectures included convolutional neural networks, Vision Transformers, and “biologically inspired” architectures with recurrent connections, and model types spanned a variety of supervised and self-supervised training objectives (see SI.2 for full list of models). We extracted model activations from several stages of each model. For ViTs we extracted activations from all intermediate encoder layers, while in all other models we extracted activations from the ReLU non-linearity after each intermediate convolutional activation. This resulted in a total number of $516$ analyzed model stages. In each case we flattened the model activations for the layer with no subsampling.*”
>
> **Minor: l 97 "self-consistently" did you mean "self-consistency"?**:
>
> This terminology refers to the fact that there is no closed-form expression for calculating $\kappa$ as a function of the model spectrum [1-3]. Instead, given fixed $\{\lambda_i\}$, we have to numerically solve for the value of $\kappa$ that satisfies the equation
> $$
> \kappa - \alpha_{reg} - \kappa \sum_i \frac{\lambda_i}{p\lambda_i + \kappa}=0.
> $$
>
> [1] Canatar et. al., Nature communications, 2021.
>
> [2] Loureiro et. al., NeurIPS, 2021.
>
> [3] Seung et. al., Phys. Rev. A, 1992.

---

> > ### Comment · Reviewer_t8VM · 2023-08-11
> > **Thanks for the clarifications**
> >
> > Dear authors, thanks for the clarifications. I have read the rebuttal.

---

> > > ### Author Response · Authors · 2023-08-18
> > >
> > > Thank you for the suggestions to improve and strengthen our paper!

---

### Official Review · Reviewer_PvxB · 2023-07-06

**Soundness:** 4 excellent
**Presentation:** 4 excellent
**Contribution:** 4 excellent
**Rating:** 9
**Confidence:** 4

**Summary:**

This study asks a crucial question: What is it about representations in ImageNet-trained neural networks that allow the successful regression of responses in mammalian visual cortex? The authors answer this question (in part) via an ingenious method combining learning theory and empirical analyses.

Even after a decade of study (or more), it is unclear why the responses of neurons in the visual cortex can be so well predicted – better than any other approach – by linearly regressing them from the representations of pretrained DNNs. Here, the authors open the door to a potentially very fruitful method of answering why. Beginning from an analytical expression for the generalization error of ridge regression, the authors notice that it can be decomposed into a product of two terms describing the geometry of the error in the space of examples (the outer product a.k.a. Gram matrix space): one term which carries the interpretation of an average radius, and the other which carries the interpretation of the effective dimensionality of the error. A low generalization error can be achieved by decreasing this radius and dimension. Then, the authors empirically (and thoroughly) analyze how the generalization error decomposes into radius and dimension when predicting V1, V2, V4, and IT responses from a very wide range of pretrained networks, both supervised and unsupervised. The results are striking. For example, networks appear to be surprisingly constrained in their “error mode geometry”, apparently forced to trade between low radius and high dimension. There are many implications in this paper for how we go about understanding neural responses.


**Strengths:**

A pleasure to review. This paper tackles a question of high significance using a thorough and deep analysis, one that is both highly original and leverages recent advances. The empirical analyses are complete and leave nearly nothing wanting.

**Weaknesses:**

The only weakness I might comment on, in the category of clarity, is the absolute density of results, many of which are not followed up or explored more deeply. To help the reader internalize these interesting results, it would be better to provide a writing structure with more summaries of the outcomes of interest both at the paragraph level and at the paper level. (I personally prefer the CCC style argued for by Mensh and Kording (2017)). At the moment the reader is left to determine for oneself which results to find interesting – meaning that to many quick readers some of these important lessons will be overlooked.

I have a number of requests for expanding the exposition. Apologies, as not all of these will fit in the page limit!

First, it would be nice to have a little more discussion and intuition of the meaning of the key error mode geometry terms, radius and dimension.

Then, there were a number of interesting findings that were only given a sentence or not mentioned in the main text.
 - SI.5.4 is very interesting but not much mentioned in the main text. Consider moving to the main text; by way of contrast this might help distinguish your own definition of dimension.
 - Line 443 in the supplement, which mentions that the eigenspectra of the stimulus set in part controls generalization error, which means that it could be a tool for better selecting stimulus sets for neural predictions. What would this look like?
 - It would be good to mention in the main text (briefly) that you analyze both supervised and unsupervised networks, and of many architectures.
 - That “the improved predictivity of trained networks in regions V2, V4, and IT is primarily driven by changes in their intrinsic expressivity as summarized by their eigenspectrum decay, rather than significant changes in their alignment with neural data.” (160-162). This would have large implications for those believing the brain==ImageNet DNNs. Could this be unpacked?

These are just a few; it would be nice to have summaries of all main takeaways.

**Questions:**

Could a plot be created in which the error mode geometry is shown split by which objective function (supervised, Barlow Twins, etc) as well as each network? Even a null result (indistinguishable) would be interesting.

Does the theory require an alignment between the eigenmodes of $G_R$ and $G_X$? If so, how is this achieved in theory?

**Limitations:**

The limitations were thoroughly mentioned and addressed. I greatly appreciated the exploration (in the supplement) of the difference between the generalization error of ridge regression and the Pearson’s R2 measure for partial least squares regression commonly used elsewhere. The additional supplement plots for other small details (outliers, train/test split, $l_\infty$ adversarial) were also appreciated. In general, I appreciate how the analyses and plots were chosen not to puff up results or push them into any particular narrative, but rather were an honest and complete presentation of the results.

---

> ### Author Rebuttal · Authors · 2023-08-10
>
> We deeply thank you for reading our paper thoroughly, your positive thoughts and detailed comments/suggestions!
>
>
> ## Weaknesses:
>
> 1. **Clarity, density of results and writing:**
>
>     We thank the reviewer for the suggestion. To make it easier for the reader to get a high level summary of our results, we’ve added a list of our main contributions to the end of the introduction (see global response).
>
>
>
> 2. **Intuition for Error Geometry: Radius and Dimensionality:**
>
>     We appreciate this suggestion and have simplified and improved the presentation of the theory (§2.2; see global response). Furthermore, we substantially revised Fig. 1 (see RFig 1) and have added a helpful schematic (RFig 2) to make our results on the error mode geometry easier to understand.
>
> 3. **SI.5.4:**
>
>     Thank you for this suggestion, and we are glad that you enjoyed this section. We will consider moving some of this to the main text if space allows. We also added the following sentence to the end of Sec. 3.1:
>
>    “*These results highlight the need to study the spectral properties of both the model and the brain responses as summarized by the $\tilde W_i$: Simply studying the properties of the eigenspectra of the model is insufficient to characterize neural predictivity (see Sec. SI.5.4 for more details).*”
>
> 4. **Line 443 in SI, tool for selecting stimulus sets:**
>
>     Thank you for this comment. In response to this, we expanded this line as:
>
>     “*These differences are not due to the recorded neurons, as the model eigenspectra is only a function of the stimulus set, pointing out the importance of considering the stimuli used for the neural predictions. Future work can consider choosing stimulus sets to maximize the difference between model eigenspectra. This would likely lead to greater differences between models in their predictions of neural activity patterns.*”
>
> 5. **Mentioning the analysis of various networks and training procedures in the main text:**
>
>     Thank you for the suggestion. We have included more details of the architecture and layer choices on lines 76-80 of the main text, and explicitly noted the use of different training objectives:
>
>     “*The neural responses and model activations were extracted for each stimulus (Fig. 1B), and each layer of each investigated neural network was treated as a separate encoding model. We examined 32 deep neural networks trained on the ImageNet classification task. Model architectures included convolutional neural networks, Vision Transformers, and “biologically inspired” architectures with recurrent connections, and model types spanned a variety of supervised and self-supervised training objectives (see SI.2 for full list of models). We extracted model activations from several stages of each model. For ViTs we extracted activations from all intermediate encoder layers, while in all other models we extracted activations from the ReLU non-linearity after each intermediate convolutional activation. This resulted in a total number of $516$ analyzed model stages. In each case we flattened the model activations for the layer with no subsampling.*”
>
> 6. **Lines (160-162) Implications for brain==ImageNet DNNs:**
>
>     Thank you very much for this suggestion! The comments around lines 160-162 were based primarily on visual inspection of differences in the respective $\{\lambda_i\}$ and $\{W_i\}$. However, in response to this comment, we ran an explicit experiment to test this claim (see RFig 4 and Global Response). To our surprise, these results indicate that training primarily drives increased predictivity through changes in alignment with the neural data, rather than changes in the eigenspectra of the model. This highlights that even if model eigenspectra seem quite different (RFig 4D), the spectral differences may not be driving the underlying differences in predictivity. We hope this serves as the groundwork for identifying the contribution of these different properties.
>
> ## Questions:
>
> 7. **Plotting error mode geometry by objective functions and networks:**
>
>     We have added this as RFig 5 and will include in the SI for V1, V2, V4, and IT. On first pass it seems like there is little difference in the error mode geometry between the supervised and self-supervised networks, which is in line with other works that suggest these networks have similar representations [1-3]. We have also added a plot grouping supervised models based on architecture (RFig 3)
>
> 8. **Alignment between $G_R$ and $G_X$ eigenmodes:**
>
>     Our theory for the generalization error depends on the alignment between the entire matrix of biological neuron responses, which is now denoted $\mathbf{G}_Y$, and the eigenvectors of the model Gram matrix $\mathbf{G}_X$, denoted by $\mathbf{v}_i$. This alignment is quantified by the weights $W_i = ||\mathbf{Y}\mathbf{v}_i||^2/||\mathbf{Y}||^2_F$, which quantifies the fraction of the variance in $\mathbf{Y}$ that the eigenvector $\mathbf{v}_i$ accounts for. As such, the eigenmodes of $\mathbf{G}_Y$ do not *explicitly* enter into the formula for the generalization error but rather $W_i = \frac{\mathbf{v}_i^\top \mathbf{G}_Y \mathbf{v}_i}{\text{Tr}\, \mathbf{G}_Y}$, i.e. its projection (see Eq. 2 in main text).
>
> [1] Geirhos et al. 2020 (NeurIPS SVRHM Workshop)
>
> [2] Zhuang et al. 2021 (PNAS)
>
> [3] Konkle & Alvarez 2022 (Nature Communications)

---

### Official Review · Reviewer_3d9A · 2023-07-06

**Soundness:** 3 good
**Presentation:** 2 fair
**Contribution:** 3 good
**Rating:** 7
**Confidence:** 3

**Summary:**

Previous works have demonstrated that many different state-of-the-art deep neural networks (DNNs) perform similarly at neural responses prediction. But a complete understanding of which aspects of these DNNs lead to the similarity in predicting neural responses remains unknown. The authors proposes a spectral theoretical framework to explore how the geometrical properties of DNNs affect the performance of neural responses prediction. More specifically, the authors introduce a generalization error to serve as a measure of fit, which brings a geometrical interpretation of the model activations (representations) in DNNs. This geometrical interpretation could further give additional insights into how DNNs are achieving different performance of neural responses prediction. The authors also design several experiments to investigate the roles of layer depth, dataset size,  and different training approaches of DNNs in neural responses prediction.

**Strengths:**

* The authors establish a novel link betwwen the predictivity of neural responses and the geometry of DNN representations.
* The authors provide a solid analysis and several comprehensive experiments to support their theoretical framework.

**Weaknesses:**

* The authors should improve the clarity of their conclusions from their results. Section 3 provides many details about their analysis and results, but it's hard for the readers to extract the essential arguments the authors want to convey. It would be better to summarize your conclusions at the beginning or end of each paragraph.
*  The authors should improve their exposition of the spectral theoretical framework. I recommend the authors include more background information and provide some examples to show the role of each equation in the relationship between the generalization error and the geometry of model activations.

**Questions:**

N/A

**Limitations:**

I have highlighted weaknesses above. I have nothing further to add here.

---

> ### Author Rebuttal · Authors · 2023-08-10
>
> We thank you for your review and suggestions. Please see our responses below.
>
> 1. **Clarity issues and summary of results:**
>
> We thank you for these suggestions and agree that the clarity should be improved. In response to this comment and others, we added a concise list outlining the principle contributions of paper to the end of the introduction. We have also internally updated our document to increase clarity of the results section focusing on the main conclusions for each section. Furthermore, we uploaded Rebuttal Figures (RFig) further explaining main concepts.
>
> 2. **Improvement for spectral theory framework:**
>
> We significantly simplified the exposition of our theory in response to this and other comments, please see the RFig 1 and RFig 2. Furthermore, please see the updated version of Sec. 2 below:
>
> > In response to a total of $P$ stimuli, we denote model activations with $M$ features (e.g. responses from one stage of a DNN) and neural responses with $N$ neurons (e.g. firing rates) by $\mathbf{X}\in\mathbb{R}^{P\times M}$ and $\mathbf{Y} \in \mathbb{R}^{P\times N}$, respectively. Sampling a training set $(\mathbf{X_{1:p}}, \mathbf{Y_{1:p}})$ of size $p < P$, the ridge regression solves: $$\hat{\beta}(p) = \arg\min_{\beta \in \mathbf{R}^{M\times N}} ||\mathbf{X_{1:p}} \beta - \mathbf{Y_{1:p}}||_F^2 + {\alpha^{reg}} ||\beta||_F^2, \quad \hat{\mathbf{Y}}(p) = \mathbf{X} \hat{\beta}(p)$$
> We analyze the neural predictivity based on the generalization error $E_g(p) = \frac{||\hat{\mathbf{Y}}(p) - \mathbf{Y}||_F^2}{||\mathbf{Y}||_F^2}$ for which we utilize theoretical tools from learning theory, random matrix theory and statistical physics to extract geometrical properties of representations based on spectral properties of data. In particular, the theory introduced in [1,2] relies on the orthogonal mode decomposition (PCA) of the Gram matrix $\mathbf{X}\mathbf{X}^\top$ of the model activations, and projection of the target neural responses onto its eigenvectors:
> >
> > $$\mathbf{X}\mathbf{X}^\top = \sum_{i=1}^P \lambda_i \mathbf{v}_i \mathbf{v}_i^\top,\quad W_i :=  \frac{||\mathbf{Y}^T\mathbf{v}_i||_2^2}{||\mathbf{Y}||^2_F}, \quad <\mathbf{v}_i, \mathbf{v}_j> = \delta\_{ij} .$$
> >
> > Here, associated to each mode $i$, $W_i$ denotes the variance of neural responses $\mathbf{Y}$ in the direction $\mathbf{v}\_i$, and $\lambda_i$ denotes the $i^{th}$ eigenvalue. Then, the predicted generalization error is given by:
> $$E_g(p) = \sum_{i=1}^P W_i E_i(p), \quad E_i(p) := \frac{\kappa^2}{1-\gamma} \frac{1}{(p\lambda_i + \kappa)^2},$$
> where $\kappa = \alpha_{\text{reg}} + \kappa \sum_{i=1}^P \frac{\lambda_i}{p\lambda_i + \kappa}$ must be solved self-consistently, and $\gamma = \sum_{i=1}^P\frac{p \lambda_i^2}{(p \lambda_i + \kappa)^2}$. Note that the theory depends not only on the model eigenvalues $\lambda_i$, but also on the model eigenvectors $\mathbf{v}_i$ along with the responses $\mathbf{Y}$, which determine how the variance in neural responses distributed among its eigenmodes.
> >
> > Although the equations are complex, the interpretations of $W_i$ and $E_i(p)$ are simple: $W_i$ quantifies the projected variance in neural responses on model eigenvectors (alignment between neural data and model eigenvectors, i.e., *task-model alignment* [1]). Meanwhile, $E_i(p)$ determines the reduction in the error contributed from each $W_i$ when the training set size is $p$, and depends only on the eigenvalues, $\lambda_i$ (i.e., *spectral bias* [1]).
> >
> > In this work, we combine both and introduce *error modes* $\tilde W_i(p) := W_i E_i (p)$:
> $$\tilde W_i(p) := \frac{\kappa^2}{1-\gamma}\frac{W_i}{(p\lambda_i + \kappa)^2}, \quad E_g = \sum_i \tilde W_i(p)$$
> As shown in RF 1C, it can be seen that $\tilde W_i$ that are associated with larger eigenvalues $\lambda_i$ will decay faster with increasing $p$ than those associated with small eigenvalues.
> >
> >While the error modes $\tilde W_i$ completely characterize the generalization performance of a given model, it is difficult to use them for direct model comparison due to their high dimensionality. Our main goal here is to derive a geometric measure that analytically relates to the generalization error. This is in contrast to previous such measures, such as the effective dimensionality that only depends on model eigenvalues.
> >
> >Here, we define a set of geometric measures that characterize the distribution of a model's $\tilde W_i$ via the ***error mode geometry*** (RFig 1D). Specifically, we rewrite the generalization error $E_g(p)$ as:
> > $$ R_{em}(p) := \sqrt{\sum_i \tilde{W}_i(p)^2}, \quad D\_{em}(p) := \frac{\big(\sum_i \tilde W_i(p)\big)^2}{\sum_i \tilde W_i(p)^2}, \quad E_g(p) = R\_{em}(p) \sqrt{D\_{em}(p)}.$$
> >
> > The error mode radius $R_{em}$ denotes the overall size of the error terms, while the error mode dimension $D_{em}$ represents how dispersed the total generalization error is across the different eigenvectors (RFig 1D). Note that, the generalization error $E_g(p)$ above has a degeneracy of error geometries; many different combinations of $R_{em}$ and $D_{em}$ may result in the same $E_g(p)$ (RFig 2)."
>
> [1] Canatar et. al., Nature communications, 2021.

---

> > ### Comment · Reviewer_3d9A · 2023-08-19
> >
> > Thank you for your efforts to address my concerns. The presentation of this paper is overall improved. I will upgrade my score to 7.

---

### Official Review · Reviewer_RZx6 · 2023-07-10

**Soundness:** 2 fair
**Presentation:** 2 fair
**Contribution:** 2 fair
**Rating:** 5
**Confidence:** 4

**Summary:**

The authors studied linear regression-based DNN encoding models of macaque visual cortical areas by extending a recent generalization error theory. They provided a theoretical link between the predictivity and geometry of representations and showed that models with similar generalization errors may have quite different geometrical properties. They compared a randomly initialized ResNet50 with standard and adversarially trained ResNet50 models. Evaluations and analyses are a bit limited.

**Strengths:**

- Analyzing the eigenspectrum of a model's feature space appears to be an interesting analysis that goes beyond the standard measures of representational similarity analysis or linear regression experiments that one usually comes across for this line of research at the intersection of cognitive neuroscience and machine learning. So, I appreciate that effort.

**Weaknesses:**

**Major**

- **Tense**: There's a mix between present tense and past perfect in $\S2$ which I think is not great. Choose one tense and use it consistently. I'd use the present tense to describe the problem settings and methods and either use present tense or past perfect for the results section. Also, use active rather than passive voice which I think is better readable (and more scientific).

- **$\S3.4$ + main results**: I find it very strange to denote the training set size with small $p$. As I mention below, $p$ is usually used to refer to the number of features in some representation space and not to the size of the (training) data. If I was just reading the title of the subsection, I'd think that the generalization error depends on the size of the feature space (although I would not know on which of the feature spaces). This is not implausible, but very different from what you mean/find. Actually, it's pretty trivial that the generalization error depends on the size of the training data for fitting the regression model. When would the generalization error not depend on the size of the training data? This section is in general a bit confusing. In lines 205-207 you write: "[...] it may be that trained networks do better than their random counterparts at predicting V1 data for large, but not small, training data set sizes." Do you refer to the size of the (training) data with which the regression model is fit? If that's the case, I am a bit puzzled because from Bayesian statistics (and statistics in general) we know that the prior is most important for small data regimes. In the infinite data limit, the prior and, therefore, any regularization parameter stops mattering. A pretrained neural network has a much better initialization compared to a randomly initialized network. It has found a good local minimum over the course of pretraining which gives it a big head start. The pertaining is an implicit regularization for any downstream task. Here, we can treat the regression task as a downstream task. So, the prior should matter most when the training data to fit your regression model is small. You seem to find the opposite, or at least that's what you claim. The only possibility where I can see this being the case is when the training data size for the downstream task (e.g., for performing linear regression) is so small that the parameters of your (regression) model can basically explain no variance at all. But then the findings are pretty useless, aren't they? Actually, I am confident that $N=135$ is too small to find any statistically meaningful effects.

- **Model choice**: It would have been interesting to see at least one, if not more, other architecture(s) being analyzed. I doubt that you can generalize any findings from a single ResNet50 to the whole space of DNNs. Even within the subclass of CNNs, there exist countless models. I find it more interesting to compare CNNs (pick two or three) to ViTs (pick two or three) but comparing a few CNNs (each with a different inductive bias --- e.g., AlexNet, VGG16, ResNet50, ResNext, ConvNeXt, EfficientNetV2), ideally trained with different recipes would have probably been sufficient.

**Minor**


- Lines 26-27: "[...] many different architecture and training procedures lead to similar predictions of neural responses." While the former is true and has been demonstrated multiple times in various ways, it has recently been shown in a comprehensive study that both training data and objective function have a major impact on the degree of alignment with human behavior; much more than architecture (see [Muttenthaler, L., Dippel, J., Linhardt, L., Vandermeulen, R. A., & Kornblith, S. (2022)](https://openreview.net/pdf?id=ReDQ1OUQR0X)). A longer while ago it has also been shown that different data augmentation strategies lead to different biases and as such to different degrees of consistency with human errors. See [Hermann, K., Chen, T., & Kornblith, S. (2020)](https://proceedings.neurips.cc/paper_files/paper/2020/file/db5f9f42a7157abe65bb145000b5871a-Paper.pdf); [Hermann, K., & Lampinen, A. (2020)](https://proceedings.neurips.cc/paper/2020/file/71e9c6620d381d60196ebe694840aaaa-Paper.pdf); [Geirhos, R., Rubisch, P., Michaelis, C., Bethge, M., Wichmann, F. A., & Brendel, W. (2018)](https://arxiv.org/pdf/1811.12231.pdf); and [Geirhos, R., Narayanappa, K., Mitzkus, B., Thieringer, T., Bethge, M., Wichmann, F. A., & Brendel, W. (2021)](https://proceedings.neurips.cc/paper_files/paper/2021/file/c8877cff22082a16395a57e97232bb6f-Paper.pdf).


- In line 58 is a small grammar mistake: "[...] characterizing how fast the eigenvalues of data Gram matrix fall [...]". There is an article missing before **the** data Gram matrix.


- **Notation**: The notation in $\S2.2$ is weird. Why is the matrix of neural activations $\mathbf{R} \in \mathbb{R}^{P \times N}$? First, $\bf{R}$ can sometimes have a special meaning. So, I'd be careful with that. Second, although not bolded, you use $R$ for the generalization error terms. Third, generally $N$ is used to denote the number of stimuli/examples in your training data and not to refer to the number of features.  My suggestion is to denote the matrix of model features as $\mathbf{X} \in \mathbb{R}^{N \times D}$ and the matrix of neural activations $\mathbf{Y} \in \mathbb{R}^{N \times P}$. As such, the regression problem $\mathbf{Y} = \mathbf{X}\mathbf{A}^{\top} + b$ becomes more intuitive. I'd replace $\mathbf{R}^{\ast}(\alpha_{reg}, p)$ with the full equation. $\mathbf{R}^{\ast}(\alpha_{reg}, p)$ is not easy to parse. That being said, note that this is just my personal taste which is why it's a minor weakness. There may be people who don't care about that. I think that maths should be easy to follow, even if a reader skims the paper. I dislike decorative maths. If it's just decorative, it can as well be omitted.


- In Equation 2 is an "is equivalent to" symbol. It is probably not clear to everyone why the two terms are equivalent but not equal. I'd like to see a (short) derivation of it. You can derive it in the Supplementary Material. Similarly, there are two "$\equiv$" symbols in Equation 5. Why did you choose "$\equiv$"? How are the terms "equivalent" but not "equal"? Do you perhaps mean "$\coloneqq$" for an assignment of variables? That would make more sense to me. "$\equiv$" seems to be an abuse of notation but maybe I am missing something here.


**Questions:**

- How do you explain a decrease in $R_{em}$ and an increase in $D_{em}$ that is caused by (standard) training?
- Why did you choose a (wide) ResNet50 and not any other model? Is there a particular reason for that choice?
- Why do you think "[...] that trained networks do better than their random counterparts at predicting V1 data for large, but not small, training data set sizes"? Is it possible that the size of the (training) data to fit the regression model is just too small to explain any variance at all or do you have another explanation that rules out that possibility? I think that $N = 135$ is too small to see any effects at all which is why I think there's no notable difference between pretrained and randomly initialized networks in your analyses. However, $N = 3600$ is large enough to observe statistically significant effects. So, I doubt that you can draw any conclusions at all about the smaller of the two datasets. Note that $N = 3600$ is still not a particularly large dataset.

**Limitations:**

Yes. The authors discussed limitations of their work in a dedicated section at the end of the main paper.

---

> ### Author Rebuttal · Authors · 2023-08-10
>
> We greatly appreciate your careful, extensive and constructive assessment of our work which led us to improve the clarity of our paper! Please also see global response and Rebuttal Figures (RFig).
>
> **Tense**: Thank you for pointing this out. We edited our draft so that the tense is consistent in each section, and we now use active voice when possible. Please see global response updates to §2.2 and §3.2 for examples.
>
> **§3.4 + main results**:
> 1. **Notation:** Thank you for pointing out this discrepancy in the literature. We use the convention where $p$ is the training set size and $P$ is the full dataset size to be consistent with previous theoretical work [1,2]. But if you feel that it is absolutely necessary, we are willing to change the notation.
> 2. **Text and Title:** We fully agree that the title and text of this section are confusing. The main purpose of this section is that metrics for "ranking" models depend on dataset size and the ordering of models can change based on this decision. This will be clarified in the main text. We also changed the section title to “*Characterizing the Dependence of Generalization Error on Training Set Size*", and updated the first sentence to “*Our spectral theory of neural predictivity characterizes the relationship between the training set size $p$, the generalization error, and the spectral properties of the model and brain responses.*”.
>
>     Note that this section mainly explains potential consequences from our theoretical results, and as such could be moved to the Discussion if you think it's appropriate.
>
> 3. **Regarding model priors and dataset size:** We apologize for the confusion and thank you for bringing up this excellent example!
>
>     While we agree with your intuition, the theory is precisely useful for such examples. In our framework, having a good prior corresponds to the variance in neural responses $\mathbf{Y}$ being fully characterized by the first few model eigenvectors $\mathbf{v}_i$ with large eigenvalues $\lambda_i$. Since the theory poses that the variance corresponding to the larger eigenvalues are learned in greater rates, the theory confirms your example when few samples are sufficient to describe the majority of the variance.
>
>     However, we observe that the distribution of the data variance among model eigenvectors on V1 data are quite similar between trained and untrained models, and widely spread. This means that in small data ($p$) regime, the explained variance remains roughly the same for both models. Then we concluded that the trained models *may* distinguish themselves in the large $p$ regime where the observed high dimensionality (hence expressivity) of the feature space ensures learning remaining portion of the variance in data.
>
> 4. **Model Choice:** Thank you for expressing concerns regarding experiments, which we believe is a confusion due to lack of experimental details in the main text. The models investigated in Fig. 2-4 spanned many different CNNs and ViT architectures. This is now clear in main text lines 76-80 (see response #5 to PvxB). We also included RFig 3 of the error mode geometry for supervised neural networks grouped into different architecture types. For Fig. 5 we used publically available checkpoints for ResNet50 so that we had different training types with the same architecture.
>
>
>
> **Minor Concerns:**
>
> 1. **Lines 26-27:** Thank you for highlighting this point and providing these references. We agree that the line of work focusing on human vs. model behavior is particularly interesting, especially given that many models have similar neural predictions. We tried to convey this in the original text Lines 27-33 (which already include some of the referenced papers), but agree that this could be further clarified. We modified the paragraph starting at line 27 and added in these citations. The final sentence of that paragraph now reads:
>
>     “*Given the increasing number of findings that demonstrate large variability among candidate computational models, it is an open question of why current neural prediction benchmarks are less sensitive to model modification, and how to design future experiments and stimulus sets to better test our models.*”
>
> 2. **Grammar Mistake in Line 58:** We updated this.
>
> 3. **Notation in §2.2:** We have changed our notation for the neural data matrix from $\mathbf{R}$ to $\mathbf{Y}$ as suggested.
>
> 4. **The $\equiv$ symbol:**  Indeed, the “equivalent” symbols meant assignment of variables. We switched to the “:=” symbol as suggested.
>
> **Questions:**
>
> 1. **$R_{em}$ and $D_{em}$ after training:** These points are now explicitly described in the revised §3.2 (see the global response), and the corresponding RFig 4.
>
> 2. **Choice of ResNet50:** Please see the response above regarding other models.
>
> 3. **Size of Training Set and V1:** We thank you for this question, which led to experiments presented in RFig 4 and RFig 6. A few things to note:
>     1. First, a similar finding has been reported before that the random networks do fairly well for early visual areas [3,4,5], and our observations that the differences are more pronounced in later regions is in line with these previous works. That said, the V1 data is still slightly better predicted by trained networks, which is now explicitly stated in RFig 4B.
>     2. Note that the V2 dataset has the same number of stimuli presented as the V1 dataset, and we see more difference between trained and random networks in this region, thus the difference cannot be only due to stimulus set size.
>     3. We have run an additional suite of experiments on V1 region data with larger data sizes ($P=1250$ stimuli) and find that our observations are consistent. Please see our response to reviewer t8VM and RFig 6 for more details.
>
> [1] Bordelon et. al., ICML, 2020
>
> [2] Canatar et. al., Nature communications, 2021
>
> [3] Schrimpf et. al., BioRxiv, 2018
>
> [4] Saxe et. al., Nature Reviews Neuroscience, 2021
>
> [5] Elmoznino et. al., bioRxiv, 2022

---

> > ### Comment · Reviewer_RZx6 · 2023-08-11
> > **Thanks for the thorough response**
> >
> > I thank the authors for taking the time to read my review and respond to it thoroughly. I am glad to see that my feedback was useful. Because I appreciate the effort that the authors made to improve their submission and I feel the responsibility to acknowledge that, I will raise my score from 3 to 5.

---

> > > ### Author Response · Authors · 2023-08-18
> > >
> > > We thank you for your thoughtful review and raising our score. We are happy to address any further concerns or questions.

---

### Official Review · Reviewer_Q3vo · 2023-07-12

**Soundness:** 3 good
**Presentation:** 2 fair
**Contribution:** 2 fair
**Rating:** 7
**Confidence:** 3

**Summary:**

In recent years, DNNs trained on image recognition tasks have emerged as strong predictive models of neural activity in the visual cortex. Typically, a regression model is trained to map DNN responses onto biological neural activity in response to the same inputs. The standard method for evaluating the predictive capacity of given DNN is to quantify the variance explained by the model on held-out images. This compresses the structure of each model into a single scalar value. While this can be used to rank models, it provides little insight into the structure of the model's representations and how they align with the structure of the biological neural representations. Another approach analyzes the similarities in representational geometry between models and neural data. However, the authors, state, these approaches do not relate the geometry of the DNN to the regression model used for the neural predictions.

In this paper, the authors leverage a recently proposed method for analyzing the generalization error of a model in terms of the radius and dimensionality of the error along each eigenvector of the input data (images viewed by both model and brain), to analyze the geometry of the generalization error of the regression model that maps a given DNN to neural data.

The authors use this method to analyze layer-wise differences in prediction error in a DNN, in terms of the error geometry. They find that the contributions of the error radius and error dimension change across layers.

They also analyze difference between trained and untrained DNNs at predicting neural responses. For V1 data, they find that the difference in predictivity and error geometry for trained vs. untrained networks is small. However, for V2 - IT data, they find interesting differences. In particular, they find that training reduces error radius, but increases error dimensionality, i.e.  spreads errors more evenly across error modes. They suggest that the improved predictivity of trained networks for V2 - IT data is primarily driven by changes in their intrinsic expressivity rather than significant changes in their representational alignment with neural data.

In addition, they examine differences between adversarially vs. standardly trained models. Replicating previous findings, they find that adversarial training overall improves model predictivity. They observe differences in how it affects error mode geometry across layers. In earlier layers, error radius is decreased with little change in error dimensionality. In later layers, error dimensionality is the main distinction between robust and standard models.

Lastly, they examine the dependence of the generalization error on the size of the training set of images. They find that for small datasets, the directions with small eigenvalues yield high error. For larger datasets, the additional samples permit learning along small eigenvalue directions.

**Strengths:**

The paper applies a recently proposed methodology for analyzing the generalization error of a model to the problem of understanding DNN-neural data representational alignment. It is a solid idea, and the authors conduct a suite of well-posed experiments that analyze existing results from the literature in this frame.

**Weaknesses:**

The paper would benefit from several changes.

First, the motivation of the methodology was unclear. Intuitively, why are we interested in understanding the radius and dimensionality of the error? What insight can this give us into understanding representational alignment? What's the motivation for projecting along the eigenvectors of the data's Gram matrix? I think strong answers to these questions can be formulated. They should appear front and center in the paper.

Second, the conclusions obtained by the experiments were difficult to tease out. These should be clearly and simply stated, and presented in a list as an overview of the results. The way the results are written and presented visually, the reader needs to do a lot of digging to parse the meaning and significance. Consider making more intuitive graphical figures to emphasize the main points / contributions / conclusions, and/or condensing the results figures into simpler plots that highlight the important trends.

Lastly, the standard approach for interpreting DNN-neural data alignment could be more clearly explained and illustrated, so the reader understands how your approach differs. Consider making a schematic plot that highlights this distinction.

Overall, I felt the paper required considerable effort to understand. Given that this work draws from several disciplines and relies on a very recently proposed methodology, I believe the authors should do more work to help the reader understand the motivation and significance of this work.

**Questions:**

I'd like to get a more intuitive handle on the significance of the error modes. I expect the eigenvectors of the gram matrix of the image data would be something like a Fourier basis. In this case, we'd expect large eigenvalues for low frequency bases and small eigenvalues for high frequency. Does this analysis essentially explain how error is distributed across frequency content in the images? If so, how does this give us insight into representational alignment?

**Limitations:**

The significance of the conclusions drawn from the experiments is not clear. The purpose of this tool is to provide insight into neural representations in DNNs and the brain. Greater effort should thus be made to make the meaning of the quantified values (error radius / dimension along eigenvalues of the data's Gram matrix) interpretable.

---

> ### Author Rebuttal · Authors · 2023-08-10
>
> We thank the reviewer for their detailed review of our work. Please see the responses below.
>
> ## Weaknesses:
>
> 1. **Motivation for methodology:**
>
>     We are thankful for the suggestions and made changes to improve the clarity of our work.
>
>
>     We found that the radius and dimensionality of the error modes are a convenient way to summarize spectral properties of the generalization error. Specifically the two measures jointly summarize whether the distribution of the error modes, $\tilde W_i$ are relatively spread out (higher $D_{em}$, lower $R_{em}$), or tightly concentrated (lower $D_{em}$, higher $R_{em}$) at a fixed generalization error. Because the total error ($E_g(p)$) is the product of these two properties, we can easily visualize how the error mode geometry of many models differs. We include RFig 2 (which we will add to the main text) to give readers additional intuition for these quantities and how to interpret our plots.
>
>
>     The motivation for projecting along the eigenvectors of the Gram matrix is that it allows us to write the generalization error in terms of the alignment coefficients, $W_i$ and the spectrum, $\{\lambda_i\}$. We have highlighted these interpretations in the revised figures (RFig 1C-D) and the revised section 2.2–see the global response for details.
>
> 2. **Conclusions for experiments and clarity:**
>
>     Thank you for this suggestion. We have added a list of key contributions to the end of the introduction (see the global response). We have significantly revised Figure 1 (RFig 1), and have expanded Figure 1E (RFig 2) to help readers interpret the contour plots. We have also replaced Figure 4B-D with RFig 4, where we performed an explicit experiment analyzing the contribution of $\lambda_i$ and $W_i$ to the generalization error to emphasize the main conclusions of this section (see global response for full details).
>
> 3. **Comparison to standard approaches:**
>
>     Our approach decomposes one commonly used measure of DNN-brain comparison (neural predictivity via regression) in terms of the spectral properties of the model and its alignment with the neural data. As such, our analyses primarily concern how and why  models achieve the predictivities that they do, rather than offering an alternative scalar metric for predictivity. Indeed, as one reviewer noted, "Analyzing the eigenspectrum of a model's feature space [....] goes beyond the standard measures of representational similarity analysis or linear regression experiments that one usually comes across for this line of research..." To make this more explicit we have added a schematic figure highlighting this aspect of our method (RFig 2).
>
> 4. **Overall:**
>
>     Thank you for this suggestion. We believe that the changes detailed in the global response have made the manuscript significantly easier to understand.
>
> ## Questions:
>
> 1. **Intuitive picture for error modes and analogy to Fourier basis:**
>
>     Please note that the Gram matrices that we refer to in the paper are defined on the artificial and biological network representations, rather than the images themselves. For example, the gram matrix $\mathbf{G}_X$ is defined as $\mathbf{G}_X = \mathbf{X}\mathbf{X}^T$, where $\mathbf{X}$ is a $P\times M$ matrix containing the activations of $M$ artificial network units responding to $P$ images. Given that the network responses have no direct connection to the frequency content of the images, neither do the Gram matrices.
>
>
>     Our work focuses on the alignment between the gram matrices of the artificial and biological network representations, denoted $\mathbf{G}_X$ and $\mathbf{G}_Y$, respectively. The alignment between these two matrices as expressed by the coefficients $W_i$ therefore quantifies the similarity between the artificial and biological network representations.
>
>
>     Nevertheless, we agree that it would be of interest to gain insight into how these error modes map onto the image space. One future direction we are interested in pursuing is visualizing error modes for a given network that are contributing the most and least to the error. It is possible that these could somewhat correspond to specific frequencies (especially when investigating early layers of the neural network). However, this type of analysis is beyond the scope of this paper.
>
> ## Limitations:
>
> 1. **Main conclusions from experiments and their significance:**
>
>     Thank you for this suggestion. To make the interpretation of $R_{em}$ and $D_{em}$ clearer to the reader, we have significantly revised the figures in the paper with helpful schematics. Furthermore, we greatly simplified the presentation of the theory in section 2.2 (see global response), and we revised section 3.2 to more clearly explain our results involving $R_{em}$ and $D_{em}$ (see global response).

---

> > ### Comment · Reviewer_Q3vo · 2023-08-11
> >
> > I thank the authors for the thorough rebuttal. The rebuttal addressed my concerns, which primarily regard the clarity of the results and presentation of the main ideas. With these clarifications incorporated into the paper, I think it will be a strong and valuable piece of work. I have adjusted my score accordingly.

---

> > > ### Author Response · Authors · 2023-08-18
> > >
> > > Thank you for your review and suggestions for how to clarify the presentation and clarity of the paper, and for thinking that this will be a strong piece of work. If you have any further questions we are happy to address them.

---

### Author Rebuttal · Authors · 2023-08-10

# Global Response

We thank the reviewers for their helpful comments and for highlighting the significance of our work. As noted by one reviewer, “Even after a decade of study...it is unclear why the responses of neurons in the visual cortex can be so well predicted...by linearly regressing them from the representations of pretrained DNNs.” Our work uses a recent theoretical framework that directly relates the generalization error from regression to the spectral bias of the model activations and the alignment of the neural responses onto the learnable subspace of the model. As noted by another reviewer, "Analyzing the eigenspectrum of a model’s feature space...goes beyond the standard measures of representational similarity analysis or linear regression experiments that one usually comes across for this line of research."

Here, we address a few concerns shared by multiple reviewers. We also upload a pdf with Rebuttal Figures (RFig).

### 1. We clarified key contributions

We added the following to the end of the introduction:

1.  We analytically decompose the generalization error of ridge regression from a model to a set of brain data in terms of the model eigenspectra, the alignment between the eigenvectors of the model and brain data, and the training set size.
2. We introduce two geometric measures that summarize these spectral properties and directly relate to the neural predictivity. We show that these measures distinguish between different models with similar predictivities using a wide variety of network architectures, learning rules, and firing rate datasets from visual cortex.
3. Using spectral theory, we demonstrate that for networks effective in predicting neural data, we can ascertain if their superior performance stems from the model's spectra or alignment with the neural data. Our findings indicate:

    (a) Trained neural networks predict neural data better than untrained ones due to better alignment with brain response.

    (b) The effect of adversarial training on these geometric measures interacts with both the cortical area and the neural network layer being analyzed.


### 2. We revised the presentation of the theory

Section 2.2 has been rewritten to improve clarity and provide intuitions. We have also updated Figure 1 and added a helpful schematic–see RFig 1 & 2. We include part of the section here (see 3d9A for the full section):

"...Although the equations are complex, the interpretations of $W_i$ and $E_i(p)$ are simple: $W_i$ quantifies the projected variance in neural responses on model eigenvectors (alignment between neural data and model eigenvectors, i.e., *task-model alignment* [1]). Meanwhile, $E_i(p)$ determines the reduction in the error contributed from each $W_i$ when the training set size is $p$, and depends only on the eigenvalues, $\lambda_i$ (i.e., *spectral bias* [1]).

In this work, we combine both and introduce *error modes* $\tilde W_i(p) := W_i E_i (p)$:
$$\tilde W_i(p) := \frac{\kappa^2}{1-\gamma} \frac{W_i}{(p\lambda_i + \kappa)^2}, \quad E_g = \sum_i \tilde W_i(p)
$$

(see Eq. SI.5 for details.) As shown in RFig 1C, $\tilde W_i$ associated with large eigenvalues $\lambda_i$ will decay faster with increasing $p$ than those associated with small eigenvalues."

The generalization performance of a model is fully characterized by its error modes, $\tilde W_i$. However, due to its vector nature, $\tilde W_i$ is not ideally suited for model comparisons. To address this limitation, we condense the overall shape of $\tilde W_i$ into two geometric measures, while preserving their direct relationship to the generalization error.

### 3. New experiment and clarification of training vs. random model results

One common concern was the clarity of results. We rewrote our results sections to  more clearly connect the spectral properties of the model and its alignment to the brain data to the error mode geometry. Additionally, we ran a new experiment to directly test whether training neural networks improves predictivity through improved alignment with neural data as opposed to changes in model eigenspectra. To demonstrate these points, we give the revised section on trained vs. untrained networks below.

"*We analyzed how the error mode geometry for neural predictions differed between trained and randomly initialized DNNs (Fig. 4A). In line with previous results [2,3,4,5], we found that training yielded an improvement in neural predictivity as measured via smaller $E_g(p)$ (RFig 4B). This improvement was most notable in regions V2, V4, and IT, where there was also a characteristic change in the error mode geometry. In these regions, while $R_{em}$ decreased with training, $D_{em}$ surprisingly *increased**.

...*To gain insight into this we further investigated how the eigenspectra ($\lambda_i$) and the alignment coefficients ($W_i$) individually contributed to the observed error mode geometry. These two spectral properties can be varied independently. We performed an experiment on the trained and random ResNet50 model activations where we measured $\lambda_i$ from one model and paired it with the $W_i$ for IT data measured from the other model (RFig 4C). When using the eigenspectra of the random model, the $D_{em}$ was lower than when using the eigenspectra of the trained model. In contrast, when using the $W_i$ terms of the random model, the $R_{em}$ was much larger than that of the trained model. This $R_{em}$ decrease when using the $W_i$ terms from the trained model is the main driver of the improvement in $E_g(p)$ when we use the trained model to predict neural data...*

*...In other words, the eigenvectors of the trained model were overall better aligned with the neural data compared to the random model.*"

[1] Canatar et. al., Nat. Comm, 2021

[2] Schrimpf et. al., BioRxiv, 2018

[3] Saxe et. al., Nat. Reviews Neuro, 2021

[4] Cadena et. al., bioRxiv, 2022

[5] Cadena et. al., NeurIPS, 2019

---

### Decision · Program_Chairs · 2023-09-21

**Decision:**

Accept (spotlight)

**Comment:**

The submission contributes theory to a paradigm dominated by correlational analysis, aiming to answer the question of what drives correlations between biological and neural network representations. The majority of reviewers praised the submission's rigor and noted the potential for broader impact beyond the neuroscience core at NeurIPS.